# Hermitian and non-Hermitian topology from photon-mediated interactions

**Federico Roccati** [1] ✉, **Miguel Bello** [2,3], **Zongping Gong** [2,3,4,5], **Masahito Ueda** [6,7,8], **Francesco Ciccarello** [9,10], **Aurélia Chenu** [1] & **Angelo Carollo** [9]

As light can mediate interactions between atoms in a photonic environment, engineering it for endowing the photon-mediated Hamiltonian with desired features, like robustness against disorder, is crucial in quantum research. We provide general theorems on the topology of photon-mediated interactions in terms of both Hermitian and non-Hermitian topological invariants, unveiling the phenomena of topological preservation and reversal, and revealing a system-bath topological correspondence. Depending on the Hermiticity of the environment and the parity of the spatial dimension, the atomic and photonic topological invariants turn out to be equal or opposite. Consequently, the emergence of atomic and photonic topological boundary modes with opposite group velocities in two-dimensional Hermitian topological systems is established. Owing to its general applicability, our results can guide the design of topological systems.

The study of topological phases and related phenomena, including edge states protected against disorder, dates back to the 1980s when the quantum Hall effect was discovered[1]. Since then, the field of topological phases of matter has expanded considerably and is now a prominent topic in contemporary condensed matter physics[2,3] and photonics[4,5]. This rapid growth was also prompted by the demand for quantum technologies that are immune to disorder and detrimental environmental interactions[6]. Currently, within the context of non-Hermitian physics, a rapidly expanding research field encompassing photonics, condensed matter, and ultracold atoms, topological invariants form a new paradigm under intense investigation[7,8].

Despite considerable research efforts in the fields of solid-state physics and photonics, the exploration of topological effects in quantum optics—and especially atom–photon interactions—remains in an early stage. Some theoretical and experimental studies used topologically protected photonic edge modes as channels facilitating

unidirectional emission[9,10], excitation/quantum state transfer between quantum emitters[11–14] and multi-mode entanglement generation[15]. Notably, for a photonic Su−Schrieffer−Heeger (SSH) model, it was predicted that an atom, being a quantum zero-dimensional defect, can seed dressed bound states that are topologically protected[16]; the essential properties and occurrence criteria for such topological dressed states were then derived on a general basis[17]. Specific investigations on topological dressed states were performed in other photonic analogs of prototypical topological models, such as the Harper−Hofstadter[18,19], the Haldane model[17,19], as well as lossy systems exhibiting non-Hermitian topology[20,21]. Furthermore, the use of atomic emission properties was proposed to sense topological phases[22]. Such investigations were motivated by recent technological advances to fabricate photonic lattices with engineered properties (large periodic one-dimensional (1D) or two-dimensional (2D) arrangements of coupled cavities/resonators) and

[1]Department of Physics and Materials Science, University of Luxembourg, L-1511 Luxembourg, Luxembourg. [2]Max-Planck-Institut für Quantenoptik, Hans-Kopfermann-Straße 1, Garching 85748, Germany. [3]Munich Center for Quantum Science and Technology, Schellingstraße 4, 80799 München, Germany. [4]Theoretical Quantum Physics Laboratory, Cluster for Pioneering Research, RIKEN, Wako-shi, Saitama 351-0198, Japan. [5]Department of Applied Physics, University of Tokyo, 7-3-1 Hongo, Bunkyo-ku, Tokyo 113-8656, Japan. [6]Department of Physics, The University of Tokyo, 7-3-1 Hongo, Bunkyo-ku, Tokyo 113-0033, Japan. [7]RIKEN Center for Emergent Matter Science (CEMS), Wako, Saitama 351-0198, Japan. [8]Institute for Physics of Intelligence, The University of Tokyo, 7-3-1 Hongo, Bunkyo-ku, Tokyo 113-0033, Japan. [9]Università degli Studi di Palermo, Dipartimento di Fisica e Chimica-Emilio Segrè, via Archirafi 36, I-90123 Palermo, Italy. [10]NEST, Istituto Nanoscienze-CNR, Piazza S. Silvestro 12, 56127 Pisa, Italy. ✉e-mail: federico.roccati@uni.lu

to coherently couple them to a set of controllable atoms/quantum emitters. For example, there have been several demonstrations in various experimental platforms such as ultracold atoms[23], circuit quantum electrodynamics[24–28], and coupled-resonator optical waveguides[29]. In these setups, the photonic lattice acts as an artificially *engineered bath*, or *environment*, for the quantum emitters as their decay rate into the lattice guided modes is larger than their decay rate in free space[30,31].

Here, a fundamental question remains unanswered: do atoms coupled to a photonic bath with known topological properties inherit any of those topological properties? If so, how are the symmetry class and topological phase of the atoms related to those of the field?

To address this issue, we adopt the standard Altland–Zirnbauer classification of topological insulators[32] and consider a general model consisting of a photonic lattice weakly coupled to a periodic arrangement of two-level emitters, such that the total system is translationally invariant (with a unit cell potentially larger than that of the bare engineered bath), see Fig. 1A. We obtain general results linking the photonic and atomic topological invariants. On the basis of the bulk–edge correspondence in the Hermitian case[33], we reveal the relationship between photonic and atomic boundary modes under open boundary conditions, and the relationship between skin modes or more general bulk anomalous dynamics in the non-Hermitian case[34,35]. The key results of this system–bath topological correspondence are summarized in Table 1.

## Results

### System–bath topological correspondence

Photons in these engineered photonic baths can mediate second-order interactions between atoms; this interaction can be described by an effective many-body atomic Hamiltonian[16] in the commonly studied regime of weak-coupling and Markovian dynamics. We investigated the topological properties of the latter Hamiltonian and demonstrated that they depend on the detuning between the atomic frequency and mean photonic frequency (typically located in the middle of the central photonic bandgap).

In the following we detail the type of systems to which our theory is applicable. To be precise, we distinguish between photonic modes of the engineered bath and those of the surrounding space. Coupling to the latter can be modeled through non-Hermitian Hamiltonians, both for the engineered bath and the quantum emitters. We may also neglect the surrounding environmental modes and use Hermitian Hamiltonians instead, assuming that the emitters couple more strongly to the engineered photonic bath modes[36,37]. Thus, the whole system is modeled by the Hamiltonian $\hat{H} = \hat{H}_e + \hat{H}_p + \hat{H}_{int}$. The free atomic Hamiltonian can be written as $\hat{H}_e = \omega_e \sum_n \hat{\sigma}_n^\dagger \hat{\sigma}_n$, where $\hat{\sigma}_n = |g\rangle_n \langle e|$. Under periodic boundary conditions, the bare photonic Hamiltonian can be expressed in terms of bath modes with a definite quasimomentum $\mathbf{k}$ as $\hat{H}_p = \sum_{\mathbf{k}\in BZ} \hat{A}_{\mathbf{k}}^\dagger H_p(\mathbf{k}) \hat{A}_{\mathbf{k}}$, where BZ represents the first Brillouin zone, $\hat{A}_{\mathbf{k}}^\dagger = (\hat{a}_{\mathbf{k},1}^\dagger, \ldots, \hat{a}_{\mathbf{k},N_b}^\dagger)$, $\hat{a}_{\mathbf{k},s}$'s are the bosonic annihilation operators of the field's normal modes, and $H_p(\mathbf{k})$ is the $N_b \times N_b$ Bloch Hamiltonian matrix (see Methods). We denote the bare resonator frequency as $\omega_0$ which we choose to be the reference energy, i.e., we set $\omega_0 = 0$. According to the standard rotating-wave approximation, the interaction between quantum emitters (QEs) and the field is described by the last term of the total Hamiltonian $\hat{H}_{int} = \sum_{n=1}^{N_c} \sum_{s=1}^{N_b} g_s(\hat{\sigma}_{ns}^\dagger \hat{a}_{ns} + \text{H.c.})$. Here, $\hat{a}_{ns}$ is the real-space annihilation operator of the resonator located in the $n$th unit cell, belonging to the $s$-sublattice ($s = 1, \ldots, N_b$), and $N_c$ is the total number of unit cells. The atomic operator, $\hat{\sigma}_{ns}$, in $\hat{H}_{int}$ has two indices to specify the resonator to which it is coupled. The coupling strength $g_s$ satisfies $g_s = g$ if a QE is coupled to the resonator $\hat{a}_{ns}$, and is set to zero otherwise. Note that $g_s$ is independent on the cell index $n$, ensuring translational invariance.

Figure 1A presents a specific instance of the studied setup. It consists of two-level quantum emitters, $N_e$ in total, with a ground state $|g\rangle$ and an excited state $|e\rangle$, which are separated by the Bohr frequency $\omega_e$. The QEs are locally coupled to a translationally invariant photonic lattice implemented by coupled single-mode resonators. The lattice

**Fig. 1 | Setup. A** Two-level quantum emitters (blue spheres) coupled to an engineered photonic bath (lattice with red spheres). The full light-atom system is translationally invariant. For concreteness, the figure shows a specific 2D example with two emitters per unit cell (light shaded area). Note how, in this case, the bare photonic lattice has a smaller unit cell (dark shaded area). Important parameters of the model are the emitter transition frequency $\omega_e$ and the light-matter coupling strength $g$. In the single-excitation sector, dissipation and decoherence due to coupling to uncontrolled environmental modes (wiggly arrows) can be modeled through the use of non-Hermitian Hamiltonians. **B** Engineered bath spectrum.

Depending on the geometry and parameters of the system, one or more (Hermitian or non-Hermitian) photonic bandgaps can emerge, i.e., photons of frequency within certain ranges cannot propagate through the engineered bath. **C** Emitter-emitter, bath-mediated interactions. If the emitters are spectrally tuned to one of the bandgaps (as shown in **B**), and weakly coupled to the environment, such that the spectral distance $\Delta$ to the nearest photonic bands is larger than the light-matter coupling, the bath degrees of freedom can be traced out, leading to effective exchange interactions among the emitters.

**Table 1 | System-bath topological correspondence**

| Spatial dimension | Hermitian topology | Non-Hermitian topology |
|---|---|---|
| 1D | $\nu_a = \nu_p$ | $\nu_a = -\nu_p$ |
| 2D | $\nu_a = -\nu_p$ | $\nu_a = \nu_p$ |
| 3D | $\nu_a = \nu_p$ | $\nu_a = -\nu_p$ |

Main results of this study. If the system and bath degrees of freedom are equal, the system topology is either preserved ($\nu_a = \nu_p$) or reversed ($\nu_a = -\nu_p$) with respect to the bath topology, according to Eq. (1). Only $\mathbb{Z}$ phases are included in the table, as topology is always preserved for $\mathbb{Z}_2$ phases.

unit cell contains $N_b$ resonators. Hence, there are (generally) as many sublattices as photonic bands.

We consider an emitter arrangement with a spatial period equal to or greater than that of the photonic lattice that is translationally invariant. This is a translationally-invariant setup, featuring no randomness (in the position of the emitters) whatsoever.

In the Markovian regime, the degrees of freedom of the bath can be traced out. When the emitters' frequency $\omega_e$ lies within a photonic bandgap and the coupling constant $g$ is small (smaller than the spectral distance between $\omega_e$ and the photonic bands), the photonic lattice then induces effective coherent interactions between the emitters, described by an effective atomic Hamiltonian $\hat{H}_a \sim \hat{H}_e + g^2 \hat{G}_p(\omega_e)$ (see Methods), where the latter is the resolvent operator of the bath[38,39], see Fig. 1.

We established a *system–bath topological correspondence* by relating the topological properties of the bare bath and those of the system dressed by the bath. At resonance, $\omega_e = \omega_0$, both the photonic and the effective emitters' Hamiltonians possess identical symmetries (see below). In addition, when the system and bath have the same number of degrees of freedom (i.e., $g_s = g$ for all $s$), we reveal *the topological preservation* and *reversal* based on the Altland–Zirnbauer (AZ) classification[32]. Specifically, the topological invariants $\nu_{a(p)}$ of $\hat{H}_{a(p)}$ are related to each other as follows:

$$\nu_a = \begin{cases} \nu_p & \text{for } \mathbb{Z}_2 \text{ phases;} \\ \nu_p(-1)^{D+\hbar} & \text{for } \mathbb{Z} \text{ phases.} \end{cases} \quad (1)$$

Here, $D$ is the spatial dimension, and $\hbar$ is 1(0) if $\hat{H}_p$ is (non-)Hermitian and belongs to the (non-)Hermitian AZ (AZ and AZ$^\dagger$) classes. In the $\mathbb{Z}$ phases, the system preserves or reverses the integer topological invariant of the bath depending on the dimension and Hermiticity. By contrast, $\mathbb{Z}_2$-protected bath phases are always inherited by the system. The proof of Eq. 1 is provided in Methods.

In addition, this result conforms to the known Hermitian–non-Hermitian correspondence[35]. The boundary mode of a $D+1$ dimensional Hermitian topological system characterized by the topological invariant $\nu$ can be mapped into a $D$-dimensional non-Hermitian system on a closed manifold, with an identical topological invariant. In fact, we demonstrate that topological preservation (reversal), i.e., $\nu_a = \nu_p$ ($\nu_a = -\nu_p$), occurs in both $D+1$ dimensional Hermitian systems as well as in $D$-dimensional non-Hermitian ones (Table 1).

The topological correspondence we found holds in the case of one emitter per resonator, namely $g_s = g$ for all $s$. In the case where there are fewer emitters than resonators ($g_s = 0$ for some $s$), this general correspondence does not hold anymore, cf. Supplementary Information.

We note that (i) our result is not limited to quantum optical systems, but holds for any open quantum system satisfying the same conditions, (ii) the photonic bath Hamiltonian can be fully general, as long as it can be written in Bloch form (see above). Also, it is important to note that the topological invariants we discuss correspond to non-interacting models, and therefore, for the atomic subsystem they only classify the single-particle/linear regime. In this sense, our results are also applicable or could be extended to classical systems (*e.g.*,

topological mechanical/acoustic systems). We now proceed illustrating one example of topological preservation and reversal in each of the four possible cases (Hermitian/non-Hermitian and odd/even spatial dimension). The proofs are detailed in Methods.

**Topological preservation and reversal**

The Bloch Hamiltonian of the entire system is

$$H(\mathbf{k}) = \begin{bmatrix} \omega_e I & gI \\ gI & H_p(\mathbf{k}) \end{bmatrix}, \quad (2)$$

where $I$ is the $N_b$-dimensional identity matrix. Remarkably, the entire atom–light Hamiltonian is topologically trivial (see Methods). The effective atomic Hamiltonian is obtained using the standard perturbation theory up to the second order (see Methods) as

$$H_a(\mathbf{k}) = \omega_e + \frac{g^2}{\omega_e - H_p(\mathbf{k})}. \quad (3)$$

The real space form, $\hat{H}_a$, is recovered using the inverse Fourier transform.

Depending on the spatial dimension, the symmetries, and whether the photonic Hamiltonian is Hermitian or not, we discovered that the topology of this effective system can either be preserved or reversed at least for the fundamental symmetry classes. Here, we present the general results for the Hermitian and non-Hermitian cases as summarized in Table 1, together with representative examples.

First we consider the Hermitian case. Altland and Zirnbauer[32] were the first to identify the 10 fundamental symmetry classes (AZ classes). The topological classification of Hermitian noninteracting systems (insulators and superconductors) was subsequently developed on the basis of their classification[40,41]. Only five of these symmetry classes are relevant for the number-conserving Hamiltonians[5], (see Supplementary Table S1). The classification is based on time–reversal symmetry (TRS or equivalently $T$), particle–hole symmetry (PHS or equivalently $C$), and chiral symmetry ($S$)[5,32,33]. Their explicit effect on the Bloch Hamiltonian $H(\mathbf{k})$ is given by $TH(\mathbf{k})T^{-1} = H(-\mathbf{k})$, $CH(\mathbf{k})C^{-1} = -H(-\mathbf{k})$ and $SH(\mathbf{k})S^{-1} = -H(\mathbf{k})$, respectively.

Both $T$ and $C$ are antiunitary operators, *i.e.*, $T = U_{\text{TRS}}K$ and $C = U_{\text{PHS}}K$, where $U_{\text{TRS}}$ and $U_{\text{PHS}}$ are unitaries and $K$ denotes complex conjugation. By applying these to $H_a(\mathbf{k})$, one can show that TRS is never broken in the following sense: $H_a(\mathbf{k})$ has TRS if and only if $H_p(\mathbf{k})$ has TRS for any atomic frequency $\omega_e$ (provided that it lies within a bandgap of $\hat{H}_p$). In turn, PHS, and therefore chiral symmetry[33], can be broken. Indeed, for $\omega_e = \omega_0 = 0$, $H_a(\mathbf{k})$ has PHS if and only if $H_p(\mathbf{k})$ has PHS, the same holding for chiral symmetry.

Conversely, in the absence of photonic symmetries, no new symmetry can be generated at the atomic level whatever the value of $\omega_e$.

This demonstrates that, on resonance ($\omega_e = \omega_0 = 0$), $\hat{H}_a$ and $\hat{H}_p$ belong to the same symmetry class; off resonance, the following transitions of symmetry classes occur when going from $\hat{H}_p$ to $\hat{H}_a$ : AIII → A, BDI → AI and D → A (we refer to the standard terminology of AZ classes, see Supplementary Table S1). Topologically distinct phases within the same symmetry class are characterized by different values of the topological invariant (*e.g.*, Zak phase, Chern number, and Chern–Simons invariants), which we generally denote as $\nu_l$, with $l = p,a$ that refer to the photonic and atomic Hamiltonians, respectively. According to the bulk–edge correspondence, $\nu_l$ represents the number of edge modes in the system under open boundary conditions, where the trivial phase has a topological invariant equal to zero[33].

To ensure that the PHS and chiral symmetry are inherited, we focus on the case $\omega_e = \omega_0 = 0$, such that $H_a(\mathbf{k}) = -g^2 H_p(\mathbf{k})^{-1}$.

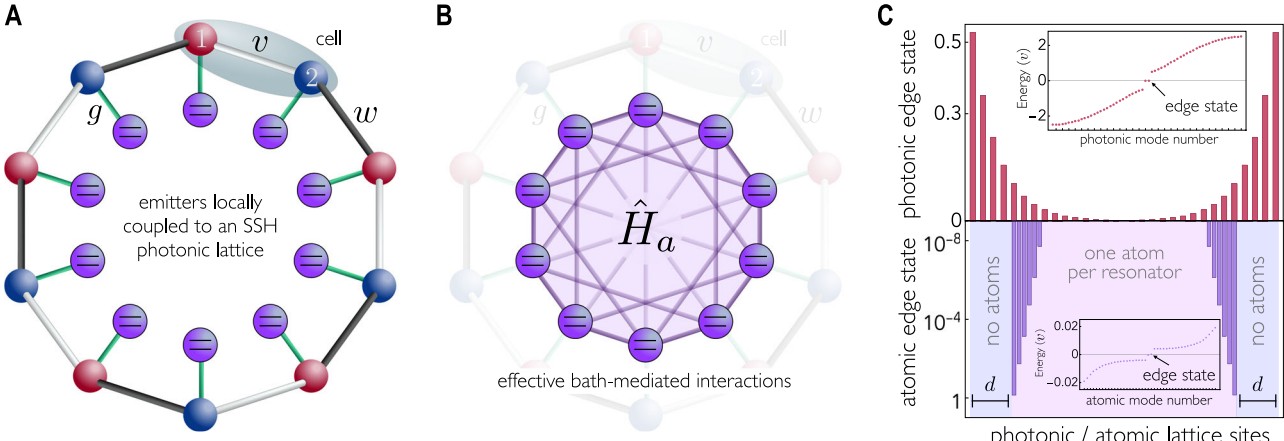

**Fig. 2 | Hermitian topological preservation. A** Scheme of the photonic Su–Schrieffer–Heeger (SSH) lattice with staggered $v$ and $w$ couplings. The coupling strength between each quantum emitter and each resonator is $g$. **B** A sketch of the mediated emitters' Hamiltonian $\hat{H}_a$ is shown in purple (where the multiple links highlight its high connectivity). Open boundary conditions for the atomic system are obtained by removing quantum emitters, but leaving the photonic lattice unaffected (hence it remains translationally invariant). **C** Modulus of the wave function of the photonic edge states with $N = 60$ resonators (top), and atomic edge states with $N_e = 44$ emitters (bottom) coupled to a periodic SSH lattice with $N$ resonators (top) and $d = 8$ sites. The resonators are numbered in increasing order including both types (1 and 2, cf. Equation 4) of resonators. Atomic open boundary conditions are obtained by removing $2d$ quantum emitters (outer violet stripes) while maintaining the periodic photonic structure. Atomic edge states are mostly localized on the first and last sites (notice the logarithmic scale). The nonzero amplitude on the remaining sites is a finite-size effect. The insets show the photonic and atomic energy spectra under open boundary conditions in units of $v$. Parameters: $w = 1.5v, g = 0.1v$, and $\omega_e = 0$.

Our main observation is that *Hermitian topology is preserved* ($\nu_a = \nu_p$) *for* $\mathbb{Z}_2$ *phases in all spatial dimensions and for* $\mathbb{Z}$ *phases in odd dimensions* (in particular 1D and 3D). Instead, $\mathbb{Z}$ *phases in even dimensions* undergo *a topological reversal* ($\nu_a = -\nu_p$) (proof given in Methods). Remarkably, the topological reversal has direct observable consequences on the basis of the bulk–boundary correspondence, as we discuss later on.

As a minimal example of 1D Hermitian topological preservation, we consider the case of QEs coupled to a Su–Schrieffer–Heeger (SSH) lattice (Fig. 2), whose Hamiltonian is

$$\hat{H}_p = \sum_n (v\hat{a}_{n,1}^\dagger \hat{a}_{n,2} + w\hat{a}_{n,2}^\dagger \hat{a}_{n+1,1}) + \text{H.c.}. \tag{4}$$

This model belongs to the BDI class, admitting $\mathbb{Z}$ phases[16]. Our hypothesis $\omega_e = \omega_0 = 0$ entails that the effective Hamiltonian between QEs preserves the chiral symmetry, and therefore, the same topology as the underlying photonic lattice based on our results, see Table 1. This is confirmed by the bulk–edge correspondence. In fact, when $\hat{H}_a$ is subject to open boundary conditions, which is indeed the case for a finite array of QEs in a larger periodic photonic SSH lattice, the effective atomic Hamiltonian supports topological edge states in the nontrivial phase despite the high connectivity of the mediated interactions.

The Hermitian reversal of topology occurs in two dimensions. We consider QEs coupled to a photonic 2D Chern insulator described by the Qi–Wu–Zhang (QWZ) model[42] (class A) to illustrate its implications. For this bipartite lattice (featuring two sublattices), the Bloch Hamiltonian $H_p(\mathbf{k})$ is

$$H_{\text{QWZ}}(\mathbf{k}) = J\sin(k_x)\tau_x + J\sin(k_y)\tau_y + J\left[u + \cos(k_x) + \cos(k_y)\right]\tau_z, \tag{5}$$

where $\tau_\alpha$ ($\alpha \in \{x,y,z\}$) are the Pauli matrices and the two fictitious spin states correspond to the two sublattices. The system is in the nontrivial phase whenever $0<|u|<2$. Assuming open boundary conditions along the $x$-direction only, the system then supports chiral photonic boundary modes propagating along $y$, in one direction

in one boundary and in the opposite direction in the other boundary. In this case, based on Table 1, topological reversal occurs so that the photonic and atomic Chern numbers have opposite sign, resulting in atomic boundary modes featuring atomic excitations that travel with opposite group velocity compared to their photonic counterparts. This is indeed the case when considering a finite array of QEs in the $x$ direction, see Fig. 3. When the set of QEs is deeply embedded in the photonic bulk, each photonic boundary mode has a corresponding atomic mode with opposite chirality on the same boundary.

We consider now the non-Hermitian case. The number of fundamental symmetry classes increases from 10 (AZ classes) to 38 (Bernard–LeClair classes) when the Hamiltonians are allowed to be non-Hermitian[43,44]. Here, we focus only on a subclass of the latter that by definition generalizes the former. In particular, complex conjugation is no longer equivalent to transposition, which is a significant difference. The non-Hermitian equivalent of the Hermitian symmetries yield the following constraints for the non-Hermitian AZ classes $U_{\text{TRS}}H^*(\mathbf{k})U_{\text{TRS}}^{-1} = H(-\mathbf{k})$, $U_{\text{PHS}}H^T(\mathbf{k})U_{\text{PHS}}^{-1} = -H(-\mathbf{k})$, and $SH^\dagger(\mathbf{k})S^{-1} = -H(\mathbf{k})$[44], where *, T, and † represent complex conjugate, transpose, and Hermitian conjugate, respectively.

Furthermore, there are the non-Hermitian AZ$^\dagger$ classes[44], for which the symmetry constraints are $U_{\text{TRS}}H^T(\mathbf{k})U_{\text{TRS}}^{-1} = H(-\mathbf{k})$, $U_{\text{PHS}}H^*(\mathbf{k})U_{\text{PHS}}^{-1} = -H(-\mathbf{k})$ and $SH^\dagger(\mathbf{k})S^{-1} = -H(\mathbf{k})$.

According to[35], the topological classification of a Hermitian AZ class in $D$ dimensions coincides with that of a non-Hermitian AZ (AZ$^\dagger$) class in $D+1$ ($D-1$) dimensions. This Hermitian–non-Hermitian correspondence is entirely consistent with the topological preservation and reversal found in this work.

For the non-Hermitian AZ and AZ$^\dagger$ classes, we find that the non-Hermitian topology is always maintained with the exception of $\mathbb{Z}$ phases in odd dimensions, where topological reversal occurs (see Methods). Here, we discuss two case studies illustrating the non-Hermitian topological reversal and preservation. We recall that the only requirement on $\omega_e$ is that it does not belong to the photonic spectrum.

Topological reversal has non-trivial consequences for 1D systems such as those considered in[20,45]. The topological origin of the largely

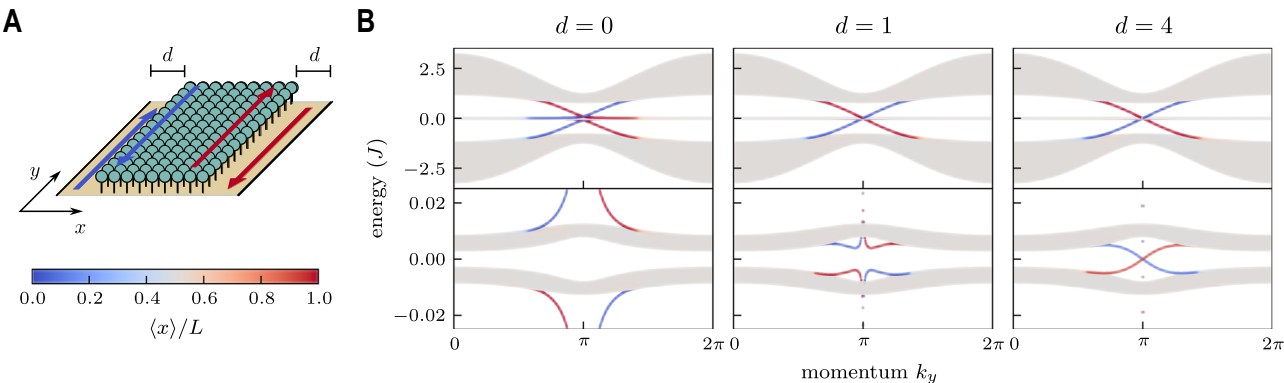

**Fig. 3 | Hermitian topological reversal. A** Quantum emitters (green spheres) coupled to a QWZ 2D photonic lattice (yellow plane); see Eq. (5). Photonic open boundary conditions are imposed only in the $x$ direction. Here, $d$ is the thickness (in photonic unit cells) of the outer stripes separating the photonic and atomic boundaries; it interpolates between the case of a system full of emitters ($d = 0$), and that of a finite array of quantum emitters in a finite but larger translationally invariant array of resonators along the $x$ direction ($d \gg 0$), at fixed $y$. We emphasize that when $d$ is sufficiently large the topological properties do not depend on it anymore. **B** Energy spectrum of the full system. Emitters are coupled to the photonic bath, except for two stripes of size $d$ along its edges. Top row: photonic spectrum. Bottom row: zoom in of the top one, displaying the atomic spectrum. The colors denote the degree of localization in the $x$ direction according to the legend (bottom left). The triviality of the full light–matter system is evident from the absence of in-gap edge states (leftmost plot, $d = 0$). The effects of the topological reversal are apparent for large $d$ (rightmost plot): at each boundary the photonic and atomic boundary modes have opposite group velocities. Parameters: $L = 50$ unit cells in the $x$ direction, $u = 1.2$, $\omega_e = 0$ and $g = 0.1J$.

investigated 1D non-Hermitian skin effect[34,46] is a point gap spectrum characterized by a nontrivial winding number $\nu$. The sign of $\nu$ can in general be related to the boundary on which the skin states accumulate.

In addition, there exist more complex symmetry-protected variants, such as the TRS[†]-protected $\mathbb{Z}_2$ skin effect in[34]. This provides an example in which the spectral winding always disappears, but a nontrivial $\mathbb{Z}_2$ number indicates the presence of two skin modes localized at both boundaries. Notwithstanding, a nonzero winding number is characteristic of nonreciprocal models and typically results from the combination of a broken TRS and dissipation[47].

When occurring, the topological reversal ensures that photonic skin states on one edge correspond to atomic skin states on the opposite edge.

Similar to[21], we consider a photonic Hatano–Nelson 1D array with nonreciprocal right $J_R = J(1 + \delta)$ and left $J_L = J(1 - \delta)$ couplings, and uniform local dissipation $\gamma = 2\delta J$

$$\hat{H}_p = \sum_n J_R \hat{a}_{n+1}^\dagger \hat{a}_n + J_L \hat{a}_n^\dagger \hat{a}_{n+1} - i\gamma a_n^\dagger \hat{a}_n \quad (6)$$

with QEs coupled to all resonators. Since only one resonator is present per unit cell, we drop the sublattice index in this discussion. Under open boundary conditions, the photonic skin modes accumulate on the right for $\delta > 0$. The atomic periodic system possesses reversed topology (opposite windings), and therefore, its skin modes accumulate to the left (Fig. 4).

The topology may be preserved in non-Hermitian cases as well. Consider a 2D chiral symmetric non-Hermitian photonic bath as an example. This system resembles the gapless surface states of three-dimensional chiral topological insulators[48,49] from the perspective of the Hermitian–non-Hermitian correspondence[35]. Accordingly, it can be characterized by the net chiral charge of Dirac cones above the base energy. In our configuration, this is equal to $\omega_e$ and is constrained by chiral symmetry to be entirely imaginary. We considered the model examined in[35]:

$$H_p(\mathbf{k}) = J\sin(k_x)\tau_x + J\sin(k_y)\tau_y + iJ\left(2\cos(k_x) + \cos(k_y) - 3\right)I_2, \quad (7)$$

where $I_2$ is the two-dimensional identity matrix. Its complex spectrum is shown in Fig. 5A. With $\omega_e$ chosen as $-iJ$, the topological invariant is 1 because there exists a single Dirac cone with a positive chiral charge located above $\omega_e$. The corresponding effective Hamiltonian for the emitters can then be derived from Eq. 3. Its spectrum is shown in Fig. 5B. The same reasoning yields a topological invariant of 1 for the emitters' Hamiltonian, indicating that the topology is preserved.

## Discussion

The above results introduce a fundamental topological reversal/preservation principle that predicts, in a universal way, the occurrence of both Hermitian and non-Hermitian topological phases of emitters effectively interacting via photon exchange. Besides its conceptual importance, our theory provides a general model-independent recipe to engineer topological phases, which can be carried out in a variety of experimental scenarios ranging from superconducting circuits and quantum optical platforms to classical oscillators and nanophotonics. Indeed, arrays of coupled superconducting resonators represent a tunable and flexible platform to implement topological photonic lattices, as it was recently demonstrated[16,26], in which case the emitters can well be implemented by superconducting resonators themselves or superconducting qubits. In addition, several recent works have demonstrated the non-Hermitian winding topology and the associated skin effect in setups such as robotic metamaterials[50], acoustic platforms[51], and topolelectrical circuits[52] (the latter in higher dimensions as well). All these setups are perfectly suited to probe the topological correspondence that we unveil, and the reversed non-Hermitian skin effect in 1D in particular.

Of course, in any physical implementation, coupling to unwanted environmental modes is inevitable, resulting in some amount of photon loss in both the emitters and the engineered bath. As a consequence, the single-particle eigenstates that we have discussed will acquire a lifetime. To observe some nontrivial dynamics this lifetime should be larger than the characteristic timescale of the bath-mediated interactions ($\sim J/g^2$). This requirement could be relaxed in the Hermitian case by post-selecting the measurements where the number of excitations is conserved. We note that the dynamics produced by the kind of bath-mediated interactions we consider have already been observed in experiments[53].

The general principle we provide predicts the topological properties of the emitters' Hamiltonian based solely on the topological properties of the photonic bath, its Hermitian/non-Hermitian nature, and its dimensionality. This system-bath topological correspondence

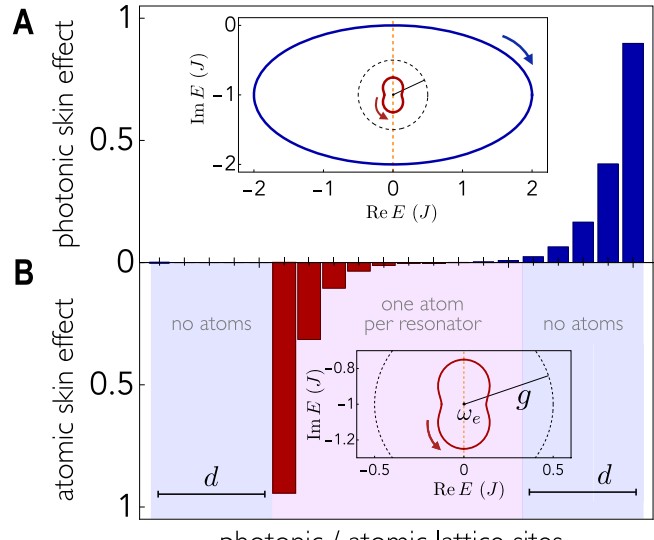

**Fig. 4 | Non-Hermitian topological reversal. A** A bare photonic 1D Hatano–Nelson model, Eq. 6 with $N = 20$ resonators, where photonic skin states accumulate on the right edge ($J_R > J_L$). When emitters are coupled to the same photonic lattice, under periodic boundary conditions, the atomic skin states accumulate on the left edge (**B**). Both figures show the normalized average of all skin modes $|\psi_i\rangle$, i.e., $\sum_i |\langle n | \psi_i\rangle|^2$, with $|n\rangle$ being the state where the photon (excitation) is located at the $n$th resonator (atom). Atomic open boundary conditions, inducing the skin effect, are obtained by removing $2d = 10$ quantum emitters (outed violet stripes) while maintaining the periodic photonic structure. The inset in A shows the photonic (blue) and atomic (red) complex spectrum under periodic boundary conditions in units of $J$. The inset in B shows a magnified view of the inset in A. We find opposite windings, witnessing the topological reversal. The vertical dashed orange axis is Re$E$ = Re $\omega_e$. The dashed black circle centered at $\omega_e$ indicates the strength of the atom–photon interaction $g$ (the radius). This is highlighted because the reversal in 1D can be described by a circular inversion with respect to this circle[57]. Parameters: $\delta = 0.5$, $g = 0.5J$, and $\omega_e = -iJ$.

sheds a new light on the emergence of exotic nonreciprocal interactions mediated by nonreciprocal 1D photonic baths observed in recent studies[20,21,45]. To showcase the effectiveness of our theory, we have considered 1D and 2D Hermitian and non-Hermitian models, thereby unveiling the occurrence of remarkable effects on the basis of the bulk-edge correspondence. For example, in a 2D Hermitian system, topological reversal enforces atomic edge modes featuring opposite group velocity compared to the photonic edge modes.

Our general classification requires that the system and the bath have the same number of degrees of freedom. By breaking this condition, we were able to show a rich variety of interesting cases (see Supplementary Information) which go beyond our topological correspondence principle. We thus provide a general criterion and fundamental hindsight that set down a cornerstone for the design of topological systems immersed in topological/non-topological Hermitian/non-Hermitian environments.

## Methods
### Photonic Hamiltonian
The photonic Hamiltonian in real space is

$$\hat{H}_p = \sum_{n,m=1}^{N_c} \sum_{s,s'=1}^{N_b} \langle \mathbf{r}_n, s | \hat{H}_p | \mathbf{r}_{n+m}, s' \rangle \hat{a}_{ns}^\dagger \hat{a}_{n+m,s'}, \quad (8)$$

where $N_c$ is the number of unit cells, $|\mathbf{r}_n, s\rangle = \hat{a}_{ns}^\dagger |\text{vac}\rangle$, and the closure relation is $\mathbb{1}_p = \sum_{n,s} |\mathbf{r}_n, s\rangle \langle \mathbf{r}_n, s|$. Because of translational invariance, the couplings are independent of the cell position, i.e., $\langle \mathbf{r}_n, s | \hat{H}_p | \mathbf{r}_{n+m}, s' \rangle = \langle \mathbf{r}_0, s | \hat{H}_p | \mathbf{r}_m, s' \rangle$.

Assuming periodic boundary conditions and using the closure relation $\mathbb{1}_p = \sum_{\mathbf{k},s} |\mathbf{k}, s\rangle \langle \mathbf{k}, s|$, where $|\mathbf{k}, s\rangle = \hat{a}_{\mathbf{k}s}^\dagger |\text{vac}\rangle$, and $\langle \mathbf{r}_n, s | \mathbf{k}, s'\rangle = \delta_{ss'} e^{-i\mathbf{k}\cdot\mathbf{r}_n}/\sqrt{N_c}$, we have $\hat{a}_{ns} = \sum_{\mathbf{k}\in BZ} e^{-i\mathbf{k}\cdot\mathbf{r}_n} \hat{a}_{\mathbf{k}s}/\sqrt{N_c}$. Thus, the photonic Hamiltonian of the periodic lattice is obtained as

$$\hat{H}_p = \sum_{\mathbf{k}\in BZ} \sum_{s,s'=1}^{N_b} \langle \mathbf{r}_0, s | \hat{H}_p | \mathbf{r}_n, s' \rangle e^{-i\mathbf{k}\cdot\mathbf{r}_n} \hat{a}_{\mathbf{k}s}^\dagger \hat{a}_{\mathbf{k}s'}. \quad (9)$$

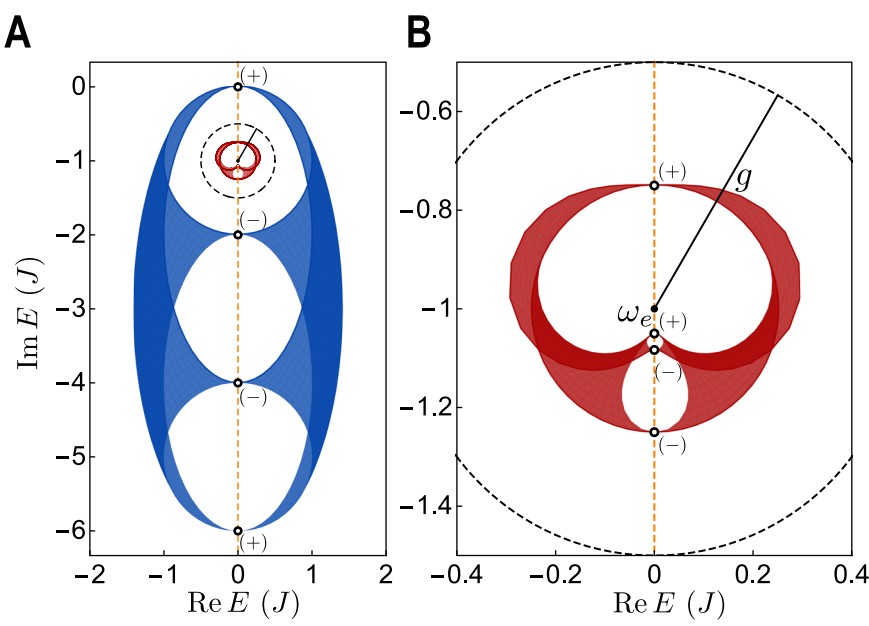

**Fig. 5 | Non-Hermitian topological preservation. A** Spectrum of the 2D chiral symmetric non-Hermitian photonic lattice[20] (blue) and spectrum of the coupled emitters (red) obtained from Eq. (3) (**B**) is a magnified image of (**A**). The dots show the Dirac cones with either (+) or (−) chiral charge. Each photonic Dirac cone above (below) $\omega_e$ is mapped to an atomic Dirac cone above (below) $\omega_e$ with the same chiral charge according to Eq. (3). The topological preservation follows from the fact that the topological invariant is equal to the total chiral charge above $\omega_e$. Dark-shaded areas are swiped twice as $\mathbf{k}$ varies in the BZ. Parameters: $\omega_e = -iJ$ and $g = 0.5J$.

By introducing the vector operator $\hat{A}_{\mathbf{k}}^\dagger = (\hat{a}_{\mathbf{k},1}^\dagger, \ldots, \hat{a}_{\mathbf{k},N_b}^\dagger)$ and denoting the matrix elements $[H_p(\mathbf{k})]_{ss'} = \sum_n \langle \mathbf{r}_0, s | \hat{H}_p | \mathbf{r}_n, s' \rangle e^{-i\mathbf{k}\cdot\mathbf{r}_n}$ Eq. (9) can be rewritten as $\hat{H}_p = \sum_{\mathbf{k}\in\text{BZ}} \hat{A}_{\mathbf{k}}^\dagger H_p(\mathbf{k}) \hat{A}_{\mathbf{k}}$, cf. Results section.

## Effective Mediated Hamiltonian

Consider the atomic frequency $\omega_e$ to be shifted by $\Delta$ from the photonic continuum. If the atom–photon coupling $g$ is weak so that $g/\Delta \ll 1$, it is possible to adiabatically eliminate the photonic bath and derive an effective photon-mediated atomic Hamiltonian $\hat{H}_a$[54,55]. The explicit expression for the case of one emitter per resonator is

$$\hat{H}_a = \hat{H}_e + \sum_{nm}\sum_{ss'} h_{ns,ms'}\hat{\sigma}_{ns}^\dagger\hat{\sigma}_{ms'}, \tag{10}$$

where

$$h_{ns,ms'} = g^2 \langle \mathbf{r}_m, s' | \hat{G}_p(\omega_e) | \mathbf{r}_n, s \rangle, \tag{11}$$

$\hat{G}_p(z) = \left(z - \hat{H}_p\right)^{-1}$ is the Green's function of the bare photonic Hamiltonian, and $|\mathbf{r}_m, s\rangle$ is the state with one excitation in the $s$th resonator of the $m$th unit cell of the photonic lattice. The double index in the atomic operators specifies both the cell $(n,m)$ and sublattice $(s,s')$ the emitter is coupled to.

As $\hat{H}_p$ is translationally invariant, so is its resolvent operator[38] and

$$\left\langle \mathbf{r}_m \middle| \hat{G}_p(\omega_e) \middle| \mathbf{r}_n \right\rangle = \frac{1}{N}\sum_{\mathbf{k}\in\text{BZ}} \frac{e^{i\mathbf{k}\cdot(\mathbf{r}_m-\mathbf{r}_n)}}{\omega_e - H_p(\mathbf{k})}, \tag{12}$$

where $\langle \mathbf{r}_m | \hat{G}_p(\omega_e) | \mathbf{r}_n \rangle$ is the $N_b \times N_b$ matrix in the sublattice space. Therefore, the atomic Hamiltonian can be written in Bloch form as $\hat{H}_a = \sum_{\mathbf{k}\in\text{BZ}} \hat{S}_{\mathbf{k}}^\dagger H_a(\mathbf{k})\hat{S}_{\mathbf{k}}$, where $\hat{S}_{\mathbf{k}}^\dagger = (\hat{\sigma}_{\mathbf{k}1}^\dagger, \ldots, \hat{\sigma}_{\mathbf{k}N_b}^\dagger)$, $H_a(\mathbf{k})$ is the Bloch Hamiltonian as in Eq. 3 and $\hat{\sigma}_{\mathbf{k}s} = \sum_n e^{i\mathbf{k}\cdot\mathbf{r}_n}\hat{\sigma}_{ns}/\sqrt{N}$, with $\mathbf{r}_n$ being the atomic operator position in real space[21]. When there are fewer emitters than resonators that are still arranged so as to preserve translational invariance, one can repeat the above arguments with a small modification: the indices $s$ and $s'$ in Eq. 11 belong only to the sublattices coupled to quantum emitters. This directly leads to the insertion of a projection operator in Eq. 12 cf. Supplementary Information.

## Triviality of the full atom-light Hamiltonian

Here, we prove that the entire atom–light Hamiltonian in Eq. (2) is topologically trivial. Its spectrum and eigenstates can be computed analytically as follows. Suppose $U_{\mathbf{k}}$ is the unitary that diagonalizes $H_p(\mathbf{k})$, $U_{\mathbf{k}}^\dagger H_p(\mathbf{k})U_{\mathbf{k}} = \text{diag}(\omega_1(\mathbf{k}), \omega_2(\mathbf{k}), \ldots, \omega_N(\mathbf{k})) \equiv \Lambda(\mathbf{k})$. Then,

$$(I_2 \otimes U_{\mathbf{k}})^\dagger H(\mathbf{k})(I_2 \otimes U_{\mathbf{k}}) = \begin{bmatrix} \omega_e I & gI \\ gI & \Lambda(\mathbf{k}) \end{bmatrix}. \tag{13}$$

Thus, for each band of the bare bath, $\omega_j(\mathbf{k})$, there are two bands $\omega_{\pm,j}(\mathbf{k}) = [\omega_e + \omega_j(\mathbf{k})]/2 \ \sqrt{[\omega_e - \omega_j(\mathbf{k})]^2/4 + g^2}$, which are the eigenvalues of $H_j(\mathbf{k}) = [\omega_e + \omega_j(\mathbf{k})]I/2 + [\omega_e - \omega_j(\mathbf{k})]\tau_z/2 + g\tau_x$.

The corresponding eigenvectors are $|v_{\pm,j}(\mathbf{k})\rangle \otimes |u_j(\mathbf{k})\rangle$, where $|u_j(\mathbf{k})\rangle$ is the eigenstate of $H_p(\mathbf{k})$ with eigenvalue $\omega_j(\mathbf{k})$, while $|v_{\pm,j}(\mathbf{k})\rangle$ is the eigenstate of $H_j(\mathbf{k})$ with eigenvalue $\omega_{\pm,j}(\mathbf{k})$. Note that regardless the value of $\omega_e$, as long as it lies in the gap of the spectrum of the bare bath, half of the spectrum is above it and half below it, i.e., $\omega_{-,j}(\mathbf{k}) < \omega_e < \omega_{+,j}(\mathbf{k})$ for all $j$ and $\mathbf{k}$. If we now consider the bands below $\omega_e$ and compute the topological invariant, we can consider instead the topologically equivalent Hamiltonian $H = I - 2P(\mathbf{k})$, where $P(\mathbf{k}) = \sum_j |u_j(\mathbf{k})\rangle\langle u_j(\mathbf{k})| \otimes |-\rangle\langle-| = I \otimes |-\rangle\langle-|$, with a constant $|-\rangle$; therefore $dH = 0$, so $\text{Ch}_n = 0$, cf. Equation (20) below.

For chiral systems in odd dimensions, the Bloch Hamiltonian of the bath can be written as[33]

$$H_p(\mathbf{k}) = \begin{bmatrix} 0 & Q_p(\mathbf{k}) \\ Q_p^\dagger(\mathbf{k}) & 0 \end{bmatrix}, \tag{14}$$

with $Q_p(\mathbf{k})$ being a suitable matrix. Then, the Bloch Hamiltonian of the bath with emitters ($\omega_e = 0$) can also be written in the same block-off-diagonal form with

$$Q(\mathbf{k}) = \begin{bmatrix} Q_p(\mathbf{k}) & gI \\ gI & 0 \end{bmatrix}. \tag{15}$$

Note that the inverse is given by

$$Q(\mathbf{k})^{-1} = \begin{bmatrix} 0 & g^{-1}I \\ g^{-1}I & -g^{-2}Q_p(\mathbf{k}) \end{bmatrix}. \tag{16}$$

Thus,

$$Q^{-1}dQ = \sum_j \begin{bmatrix} 0 & 0 \\ g^{-1}\partial_j Q & 0 \end{bmatrix} dk_j \tag{17}$$

As a consequence, $\text{Tr}[(Q^{-1}dQ)^{2n+1}] = 0$, so $\nu_{2n+1} = 0$, cf. Eq. (19).

Finally, we provide an alternative proof showing that the entire system is trivial without referring to any formulas of topological invariants. Equation (2) can be continuously deformed into $H_1 = (\omega_e I_2 + g\tau_x) \otimes I$, with $I_2$ being the $2 \times 2$ identity matrix, via a linear interpolation $H_\lambda(\mathbf{k}) = (1-\lambda)H(\mathbf{k}) + \lambda H_1, \lambda \in [0,1]$.

Further, $\det(H_\lambda(\mathbf{k}) - \omega_e I_2 \otimes I) = \det(-g^2 I) \neq 0$, so the Hamiltonian remains gapped near $\omega_e$ during the deformation. Note that any time–reversal symmetry is preserved, and so is the particle–hole (chiral) symmetry if it is extended as $(-C) \oplus C((-S) \oplus S)$. Since $H_1$ does not depend on $\mathbf{k}$ and is thus trivial, we conclude that $H(\mathbf{k})$, which is continuously connected to $H_1$, is also trivial. Note that the above proof applies equally to Hermitian and non-Hermitian systems. Moreover, the fact that an appropriately extended chiral symmetry requires a minus sign on the emitter side explains why the triviality of the entire system does not contradict the topological preservation in chiral symmetric systems.

## Proof of topological preservation and reversal

Here, we provide a general analysis of the fundamental symmetry classes (in Hermitian AZ, non-Hermitian AZ, or non-Hermitian AZ$^\dagger$) that exhibit topological reversal or otherwise topological preservation for the one-emitter-per-resonator setup. To ensure that the PHS and chiral symmetry are inherited, we focused on the case of $\omega_e = \omega_0 = 0$ ($\omega_0$ is the bare resonator frequency), so that $H_a(\mathbf{k}) = -g^2 H_p(\mathbf{k})^{-1}$ with both $H_a(\mathbf{k})$ and $H_p(\mathbf{k})$ gapped near 0. We observed that for the non-Hermitian case, both Bloch Hamiltonians must be point-gapped around $\omega_e$ with a negative imaginary part so that their spectra lie below the real axis in the complex energy plane. This is a rigid shift along the imaginary axis that does not affect the following discussion.

One obvious observation is that the mapping from $H_p(\mathbf{k})$ to $H_a(\mathbf{k})$ is invertible. This immediately implies that after obtaining the topological equivalence classes of $H_a(\mathbf{k})$ and $H_p(\mathbf{k})$, we obtain an automorphism on the classification group. Recalling that nontrivial Hermitian AZ classes are classified by $\mathbb{Z}_2$ or $\mathbb{Z}$, and so are the non-Hermitian AZ (AZ$^\dagger$) classes, it suffices to consider the automorphisms on $\mathbb{Z}_2$ or $\mathbb{Z}$ (with respect to addition). We note here that in the literature, the topological classifications of some classes are usually denoted as $2\mathbb{Z}$, meaning that the winding number or Chern number can only be an even integer. Nevertheless, since $2\mathbb{Z}$ is isomorphic to $\mathbb{Z}$, the convention $\mathbb{Z}$ is also used. In the former case ($\mathbb{Z}_2$), the only automorphism is the identity map, implying that all the $\mathbb{Z}_2$ phases exhibit topological

preservation. In the latter case ($\mathbb{Z}$), the only two possibilities of an automorphism are the identity map and inversion ($n \mapsto -n$), corresponding to topological preservation and reversal, respectively. We emphasize that the above results apply to both Hermitian and non-Hermitian systems. The problem is thus reduced to distinguishing the $\mathbb{Z}$ phases exhibiting topological reversal from those exhibiting topological preservation.

We first consider the Hermitian case. Using the band flattening ($H \to \mathrm{sgn} H$) technique[41], the map from $H_p(\mathbf{k})$ to $H_a(\mathbf{k})$ can be simplified into a simple inversion ($H \to -H$) at the level of topological equivalence classes. If the spatial dimension is odd, all the $\mathbb{Z}$ phases are chiral symmetric, and thus, the Bloch Hamiltonian takes the following form:

$$H(\mathbf{k}) = \begin{bmatrix} 0 & Q(\mathbf{k}) \\ Q(\mathbf{k})^\dagger & 0 \end{bmatrix}. \tag{18}$$

The integer topological invariant is the winding number given by

$$\nu \propto \int_{BZ} \mathrm{Tr}\left(Q^{-1}dQ\right)^D. \tag{19}$$

Obviously, this topological invariant does not change under inversion $H \to -H$ (leading to $\to -Q$). Otherwise, in even spatial dimensions, the topological invariant is the Chern number, which is determined by the flattened Bloch Hamiltonian $H(\mathbf{k})$ via

$$\mathrm{Ch} \propto \int_{BZ} \mathrm{Tr}\left(H(dH)^D\right). \tag{20}$$

However, unlike the winding number, the Chern number is inversed upon the inversion of the Hamiltonian.

We now move to the non-Hermitian case. Here, the counterpart of band flattening is unitarization[56], $H \to V = H\left(\sqrt{H^\dagger H}\right)^{-1}$, upon which the photon–atom map is simplified to $V \to -V^\dagger$. The topological invariants in odd dimensions are the winding numbers given in Eq. 19 and we always have a topological reversal. In even dimensions, any $\mathbb{Z}$ topological phase exhibits a chiral symmetry $S$, i.e., $SH^\dagger(\mathbf{k})S^{-1} = -H(\mathbf{k})$. The integer topological invariant is then given by the Chern number, Eq. 20 for $H = iSV$[44], which is confirmed to be Hermitian and flattened (i.e., square to identity). After the operation $V \to -V^\dagger$, this quantity turns out to undergo a unitary conjugation, $iSV \to -iSV^\dagger = iVS = S^{-1}(iSV)S$, leaving the Chern number unchanged.

In summary, for the Hermitian AZ classes, a topological reversal occurs only for $\mathbb{Z}$ phases in even dimensions. For the non-Hermitian AZ and AZ$^\dagger$ classes, a topological reversal occurs only for $\mathbb{Z}$ phases in odd dimensions.

## Data availability
All data needed to evaluate the conclusions in this study are present in the paper and in the Supplementary Information.

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

## Acknowledgements

We acknowledge Diego Porras and Pablo Martínez-Azcona for useful discussions. F.R. and A. Ch. acknowledge support from the Luxembourg National Research Fund (FNR, Attract grant 15382998). M.B. was supported by the projects FermiQP and EQUAHUMO of the Bildungsministerium für Bildung und Forschung (BMBF). Z.G. was supported by the Max-Planck-Harvard Research Center for Quantum Optics (MPHQ). M.B. and Z.G. acknowledge financial support from the Munich Center for Quantum Science and Technology (MCQST), funded by the Deutsche Forschungsgemeinschaft (DFG) under Germany's Excellence Strategy (EXC2111-390814868). M.U. acknowledges the support by KAKENHI Grant No. JP22H01152 from the Japan Society for the Promotion of Science. A.Ca. and F.C. acknowledge support from European Union – Next Generation EU through Project Eurostart 2022 Topological atom-photon interactions for quantum technologies (MUR D.M. 737/2021) and through Project PRIN 2022-PNRR no. P202253RLY "Harnessing topological phases for quantum technologies".

## Author contributions

F.R., M.B. and A.Ca. initiated the idea. F.R., M.B. and Z.G. developed the theory. All authors discussed the results and contributed to writing the manuscript. M.U, F.C. and A.Ch supervised the work.

## Competing interests

The authors declare no competing interests.
