## [Peer review file · Nature Communications]

Reviewers' comments:

Reviewer #1 (Remarks to the Author):

Topological quantum optics involves a topological quantum system including quantum emitters. Such systems have rich exotic physics. The authors study topological preservation and reversal, topological invariants, and topological boundary modes in Hermitian and non-Hermitian cases. Generally, the manuscript is very hard to follow because the writing and organization are not coherent. The models are messy. The results are mixed. The major problems are following:

1) The "bath" is important throughout the investigation. However, it is unclear what is the bath. A bath or environment typically means an open space to which quantum emitters or field modes decay. From system figures, I can't see what is the bath. According to the text, I can only guess the authors refer to the resonators as the bath. This is not correct, in particular, the authors only consider a single mode for each resonator. How does the bath interact with the quantum emitters?

2) The authors present a specific setup in Fig. 1B but a general Hamiltonian. The Hamiltonian does not match the setup, in which two QEs are included in each unit cell and each QE only couples to one resonator.

3) The authors consider a periodic resonator array but claim that the quantum emitter randomly couples to the resonators in each unit cell, according to the third paragraph in page 4. From the context, the Hamiltonian Hint means that the number of quantum emitters can be different in each unit cell. This will break the periodic property. Then, it is unclear how the authors discuss the topology of such system when the period breaks.

4) The authors first define the Hamiltonian H_p in the second paragraph in page 3. Then, they present another definition for H_p in page 4. What is the difference between these two Hamiltonians?

5) It is unclear what is the real form of $H_p(K)$. It is important for readers to understand the paper to present the Hamiltonian $H_p(K)$.

6) This manuscript presents three very different topological systems. However, the authors try to explain or model them with one uniform Hamiltonian? I can't understand the logic. In Fig. 2A, the QEs couple to each other besides resonators. These couplings between QEs are missed in the model.

7) Normally, QEs and resonators decay to bath. It is unclear for me what Hermitian and non-Hermitian cases mean. It means the cases $\gamma=0$ and $\gamma\neq 0$?

8) Sometimes N means the total number of unit cells, sometimes it indicates the number of resonators. This is messy and makes the manuscript unreadable.

Besides, the current manuscript includes red texts in discussion and the reference part. It looks like a draft but not a completed manuscript. Because of the aforementioned problems, I can't recommend a publication of this manuscript.

Reviewer #2 (Remarks to the Author):

In this paper, the authors study a class of topological photon-atom lattices and reveal an underlying connection between the topology of the photonic and the atomic Hamiltonians. This complements the authors' prior work, wherein they devised methodologies to induce a non-trivial topological order in lattices of quantum emitters through interactions mediated by Hermitian and non-Hermitian photonic baths. Here, they generalize their results, to create a classification for both Hermitian and non-Hermitian cases that predicts a reversal or preservation of topological invariants, between the photonic and the atom-photon Hamiltonians, based on symmetries and dimensionality. The results are interesting and indeed provide criteria for predicting the topology of certain photon-bathed atomic lattices. However, their generality is questionable, especially considering the authors' assertion of "providing general theorems on the topology of photon-mediated interactions" as stated in their abstract. Overall, it appears that the applicability of the scheme is constrained by specific approximations, selected parameters and most importantly relies on strict rules for the way atom emitters can be arranged on the photonic resonator lattice, as revealed by the scope of cases studied. In that regard, while the technical aspects are correct, I do not think this work meets the scope of Nature Communications.

According to the manuscript, the atom emitter must conform to equation 3 which, as the authors imply, requires a large separation between the atomic frequency and the photon spectrum. Moreover, the proof presented in section D is limited to $\omega\varepsilon=0$. It is apparent that for non-zero values the combined spectrum will lose all symmetries (minus TRS). Similarly, in the non-Hermitian cases the authors select $\omega\varepsilon=-iJ$, a value that sets the atom spectrum at presumably an optimal position to preserve the topology of the bath. I believe the authors should emphasize more the reliance of their scheme and its sensitivity to these parameters. Will the classification fail for different values of atomic frequencies? In a more complex scenario, can the emitters have different frequencies on the same lattice? What will happen to

the topology in such case? Can it be preserved on a sublattice where $\omega\varepsilon=0$ (for the Hermitian case) if the energy separations with other atoms are generally large?

In Fig. 3 it seems that an optimal range for the number of missing atoms exists. For $d=0$ the authors prove the combined lattice is trivial, while $d=4$ seems ideal. What dictates the selection of d and how is it connected to the topology? Are there general rules? What happens for $d>4$?

In Fig. 2, it is not clear if the photonic lattice must remain periodic (as panel 'A' and the text implies) or OBCs are also applied (as implied by the photon spectrum shown to the right). Moreover, edge states are shown for $d=8$, in which case the atomic lattice terminates 'properly' according to the OBC rules for an SSH Hamiltonian. In that case, should we expect the absence of edge states for $d=7$ (trivial topology)?

An important aspect of topological transport is its inherent resilience against disorder. To what extent these models exhibit protected transport, given that the topological properties depend on $\omega\varepsilon$ and d ?

While it may escape the scope of this work, it would be interesting to speculate whether analogous principles can be applied to Floquet topological baths by expanding the number of symmetry classes.

Reviewer #3 (Remarks to the Author):

The manuscript titled "Hermitian and Non-Hermitian Topology from Photon-Mediated Interactions" by F. Roccati et al. presents a general theorem regarding the topology of a quantum emitter (QE) array coupled to either Hermitian or non-Hermitian photonic bath with specific topological properties. The authors consider the total Hamiltonian of the QE-bath (photonic lattice) system and subsequently derive the effective Hamiltonian of the QEs by tracing out the degrees of freedom of the bath. This approach allows for the treatment of photonic lattices as mediators, facilitating photon-mediated interactions between QEs. The authors' findings reveal that the topology of this effective system can either be preserved or reversed across fundamental symmetry classes.

This photon-mediated interaction mechanism and the approach of effective Hamiltonian were previously discussed and implemented in their previous work [43], *Optica* 9, 565 (2022). In this previous work, they discussed how non-Hermitian and topological baths could induce non-reciprocal and nearest-neighbor couplings between emitters, a key feature of the topological Hatano-Nelson array. On the other hand, this bath-engineering and effective Hamiltonian approach have been implemented in non-Hermitian photonic platforms, such as parity-time (PT) symmetric and anti-PT symmetric dimmers, in both classical and quantum optics, such as *Optics Letters* 43 (21), 5371 (2018), *Nature Communications* 8, 1909 (2017),

Quantum 7, 982 (2023). However, it's important to note that the baths in these works were generally Hermitian and lacked topological properties.

This manuscript bridges the gap between the topological properties of the bath and the "system" (in this case, the QE array), providing answers to the questions raised in the manuscript, particularly regarding whether atoms coupled to a photonic bath with known topological properties inherit any of those topological characteristics. I found the results presented in the current manuscript to be important and interesting. This photon-mediated interactions introduced in this study has the potential to guide the experimental design of topological emitters across various platforms. Therefore, I recommend this work for publication in Nature Communications.

Here are some minor points that the authors should address in a revised manuscript:

1. It would be helpful to add a schematic figure to illustrate the total atom-bath system (similar to Fig. 1b) and the effective atomic array with photon-mediated interaction (similar Fig.2a).
2. In Eq. (1), η is 1 if H_p is Hermitian, and 0 if H_p is non-Hermitian. Please modify the sentence " η is 1(0) if H_p is (non-)Hermitian" to clarify that this is only for Hermitian AZ classes and non-Hermitian AZ and AZ^\dagger classes to prevent potential confusion, as a general non-Hermitian system can continuously transition to a Hermitian one. For instance, in a Hatano-Nelson 1D array, the coupling defieerence δ can continuously change from nonzero (non-Hermitian) to zero (Hermitian).
3. Please clarify the proof of Eq. (1) can be found in supplementary section S2D .
4. In the first paragraph of Non-Hermitia case, please clarify that $*$, T , \dagger represent complex conjugate, transpose, and Hermitian conjugate, respectively.
5. Below Eq. (6), "with QEs coupled to all modes", here "modes" should be "resonators" or unit cells.
6. Please specify I_2 in Eq. (7) is the 2-dimensional identity matrix.

Reply to all the Reviewers

We thank all the Reviewers for refereeing our manuscript and we are grateful for their comments and constructive critiques. We have carefully considered all the Reviewers' comments, provided a point-by-point reply (below, with quoted text in italic and our replies in blue) and accordingly modified our manuscript. All the new/modified text in the manuscript is in red.

Sincerely,

Federico Roccati, Miguel Bello, Zongping Gong, Masahito Ueda, Francesco Ciccarello, Aurelia Chenu, Angelo Carollo

Reply to Reviewer#1

We thank the Reviewer for the comments and the interesting questions. Below we carefully reply to the Reviewer's concerns, which we hope to have fully addressed

Topological quantum optics involves a topological quantum system including quantum emitters. Such systems have rich exotic physics.

We are glad to notice how the Reviewer recognizes the richness of the physical setups we consider.

The authors study topological preservation and reversal, topological invariants, and topological boundary modes in Hermitian and non-Hermitian cases. Generally, the manuscript is very hard to follow because the writing and organization are not coherent.

We thank the Reviewer for the comment and we would like to reply by describing the logic of the organization of our manuscript.

Our general result depends on two properties: Hermiticity and spatial dimensionality. Importantly, it is not relegated to quantum optical systems only, but generally to a system coupled to a topological bath.

Therefore, we decided to structure the **RESULTS** section in the following way.

First, we illustrate the System-Bath Topological Correspondence, by describing the quantum optical system, the model and the assumptions, the result, and finally specifying the validity of this result for other kinds of quantum systems. We decided to illustrate the general result leaving the proof to the Supplementary Information to ease the reading. For this reason, we only illustrate examples instead of writing tedious mathematical proofs in the main text.

At this point the manuscript necessarily becomes "two dimensional", in the sense that our result depends on two properties: Hermiticity and spatial dimensionality. We then show our examples first in the Hermitian case (both in one and two spatial dimensions, to illustrate topological preservation and reversal, respectively) and second in the non-Hermitian case (both in one and two spatial dimensions, to illustrate topological reversal and preservation, respectively).

We then conclude with our discussion, leaving all the details in the Supplementary Information.

The models are mess. The results are mixed.

We apologize, but we are not sure we understand what the Reviewer means with this statement.

The models for the photonic environments we use for our examples are the simplest toy models for each case (Hermitian/non-Hermitian and/or one/two-dimensional), used as textbook minimal models, see for example the book on the subject: Asbóth, János K., László Oroszlány, and András Pályi. *A Short Course*

on Topological Insulators. Vol. 919. Lecture Notes in Physics. Springer, 2016 and the topical review E. J. Bergholtz et al., *Rev. Mod. Phys.* 93, 015005 (2021).

It seems that there might be a misunderstanding. Our results are theorems that apply to a wide range of different systems, and are therefore formulated in general terms without fully specifying the bath Hamiltonian. To stress this fact we have added (in red) the following sentence in the new manuscript

“We note that [...] (ii) the photonic bath Hamiltonian is fully general, as long as it can be written in Bloch form (see above).”

at the end of the **System–Bath Topological Correspondence** section.

The manuscript is purposely lay down in what we believe to be a highly pedagogical structure, by conveying our messages mostly through examples, which showcase topological reversal and preservation for every significant scenario. The full-fledged rigorous proofs of our statements, together with some counterexamples, are relegated to the Supplementary Information to avoid unnecessary clutter and distraction from the main message.

Nevertheless, we made quite a few changes to further ease the reading. Namely, we fully restructured the beginning of the **RESULTS** part [see below our reply to the question 4) by the Reviewer].

The major problems are following:

1) *The “bath” is important throughout the investigation. However, it is unclear what is the bath.*

The bath is the photonic environment which the quantum emitters are coupled to. This is written in the motivating question, and at the beginning of the **System–Bath Topological Correspondence** section. Specifically, the photonic bath Hamiltonian is what we call throughout the paper \hat{H}_p (2nd paragraph of **System–Bath Topological Correspondence**). A sketch of the total system (QEs, photonic bath and their interaction) is also displayed in the new Fig. 2.

A bath or environment typically means an open space to which quantum emitters or field modes decay. From system figures, I can't see what is the bath. According to the text, I can only guess the authors refer to the resonators as the bath. This is not correct, in particular, the authors only consider a single mode for each resonator.

We respectfully disagree. Nowadays it is possible to experimentally interface quantum emitters with photonic arrays of coupled resonators so that the latter act as an environment for the emitters as their decay rate into these guided modes is much larger than their decay rate in free space [cf., A. S. Sheremet et al., *Rev. Mod. Phys.* 95, 015002 (2023)]. In the caption of each figure, the photonic lattice is the bath. Indeed we consider a single mode per each resonator. From the theoretical standpoint, when QEs are coupled to a thermodynamically large number of resonators, the latter indeed acting as an effective environment/bath for the QEs [see, e.g. the following refs where the term “bath” is used in an analogous context: T. Shi et al., *Phys. Rev. X* 6, 021027 (2016); A. González-Tudela and J. I. Cirac. *Phys. Rev. Lett.* 119, 143602 (2017)].

How does the bath interact with the quantum emitters?

The quantum emitters interact with the bath through the interaction Hamiltonian \hat{H}_{int} that appear at the end of the first paragraph of the “Results” section in second page. We sincerely do not understand what the Reviewer is asking here. Indeed, at point 3), the Reviewer seems aware that \hat{H}_{int} represents the way in which the bath is interacting with the emitters.

2) *The authors present a specific setup in Fig. 1B but a general Hamiltonian. The Hamiltonian does not*

match the setup, in which two QEs are included in each unit cells and each QE only couples to one resonator.

We regret if this point may cause confusion. The schematics in Fig 1B is meant to display a specific (though general) configuration and exemplify the possible relation between number of emitters and number of resonators per unit cells, even though we confine ourselves to the case of one emitter per lattice site. To stress this fact, in the caption we specified:

“The figure shows a specific one-dimensional example with two emitters in a photonic unit cell with three resonators (red, green, and blue), for clarity”

3) *The authors consider a periodic resonator array but claims that the quantum emitter randomly couple to the resonators in each unit cells, according to the third paragraph in page 4.*

Throughout the paper, we never write that the emitters are randomly coupled to the resonators. Indeed, in the 4th paragraph of the **System–Bath Topological Correspondence** section we state:

“We consider an emitter arrangement with periodicity equal to or greater than that of the photonic lattice that is translationally invariant.”

To clarify that the QEs are not randomly coupled we added the following sentence (in red in the new version of the manuscript):

“This is a translationally-invariant setup, featuring no randomness (in the position of the emitters) whatsoever.”

From the context, the Hamiltonian H_{int} means that the number of quantum emitters can be different in each unit cells. This will break the periodic property.

We thank the Reviewer for raising this possibly confusing point. From the form of the interaction Hamiltonian (“ $\hat{H}_{\text{int}} = g \sum_{n=1}^{N_c} \sum_{s \in \mathcal{C}} (\hat{\sigma}_{ns}^\dagger \hat{a}_{ns} + \text{H.c.})$, where $\mathcal{C} \subseteq \{1, \dots, N_b\}$ ”), since the set \mathcal{C} does not depend on the cell, *i.e.*, on n , the number of QEs per unit cell is the same in each unit cell. Therefore, translational invariance is not broken. To make this more clear we added the sentence:

“(which does not depend on the specific cell so to preserve translational invariance)”

in the 2nd paragraph of the **System–Bath Topological Correspondence** section.

Then, it is unclear how the authors discuss the topology of such system when the period breaks.

We note that we never discuss topology when periodicity breaks as (i) it would be meaningless in this context and (ii) periodicity is never broken in our manuscript (under periodic boundary conditions).

4) *The authors first define the Hamiltonian H_p in the second paragraph in page 3. Then, they present another definition for H_p in page 4. What is the difference between these two Hamiltonian?*

We thank the Reviewer for noticing this possibly misleading point. Indeed there is no difference. In the new version of the manuscript, there is now only one section (**System–Bath Topological Correspondence**). This section now introduces the quantum optical setup, discusses the result, and highlights the fact that the latter holds for any quantum system under similar assumptions. We added this sentence in red at end of **System–Bath Topological Correspondence** section:

“We note that (i) our result is not limited to quantum optical systems, but holds for any open quantum system satisfying the same conditions, [...]”

5) *It is unclear what is the real form of $H_p(K)$. It is important for readers to understand the paper to present the Hamiltonian $H_p(K)$.*

We thank the Reviewer for noticing this point. If by “real form” the Reviewer means the explicit expression of $H_p(\mathbf{k})$, then this is not relevant. Our result does not depend on the specific form of $H_p(\mathbf{k})$, supporting its generality. If instead by “real form” the Reviewer means the explicit expression of the photonic Hamiltonian in real space, then we agree that it might be obscure. However, the key point is exactly the following: the form of the photonic Hamiltonian in real space does not matter, as long as it can be written in Bloch form $H_p(\mathbf{k})$. Indeed, we prove our results by making use of $H_p(\mathbf{k})$. To stress this fact we have added (in red) the following sentence in the new manuscript

“We note that [...] (ii) the photonic bath Hamiltonian is fully general, as long as it can be written in Bloch form (see above).”

at the end of the **System–Bath Topological Correspondence** section. Nevertheless, the photonic Hamiltonian in real space is written in Eq. S6 in the Supplementary Information.

6) *These manuscript presents three very different topological systems. However, the authors try to explain or modelling them with one uniform Hamiltonian?*

We apologize, but we are not sure we understand this question by the Reviewer. In our manuscript we provide four examples of our general result, which indeed are modelled in a similar way.

I can't understand the logic. In Fig. 2A, the QEs are couples to each other besides resonators. These coupling between QEs is missed in model.

We stress that the QEs in our manuscript are not directly coupled to each other. As we write in the caption of Fig. 3A, “A sketch of the *mediated* emitters' Hamiltonian is shown in purple”. Therefore, the links between the emitters are not direct couplings, rather are the couplings mediated by the photonic bath. We have now added the new Fig. 2, showing a sketch of the general setup made of QEs (not directly coupled to each other) coupled to a photonic bath, and the effective QEs' photon-mediated interactions when the photonic bath is traced out.

In addition, we note that the Hamiltonian \hat{H}_e features no term describing a direct emitter-emitter interaction.

7) Normally, QEs and resonators decay to bath. It is unclear for me what Hermitian and non-Hermitian cases mean. It means the cases $\gamma=0$ and $\gamma\neq 0$?

In the full Hamiltonian, the bare Hamiltonian of the QEs and the \hat{H}_{int} are always assumed to be Hermitian. It is only the Hamiltonian of the photonic bath which can either be Hermitian or not. Notice, however, that the (non-)Hermiticity of the bath implies the (non-)Hermiticity of the effective Hamiltonian of the photon-mediated interaction between the QEs. We agree that a typical way of realizing a non-Hermitian evolution is by assuming a dissipative process. We also like to think of this as one of the most physical pictures in our Quantum Optics applications, although other cases are possible. One of this is for example a non-reciprocal coupling between neighboring resonators, as in the Hatano-Nelson model [which can be implemented through engineered dissipation, see, *e.g.*, A. Metelmann and A. A. Clerk. *Phys. Rev. X* 5, 021025 (2015); K. Fang et al., *Nature Phys* 13, 465–471 (2017);]. Essentially, the only assumption we make is that the spectrum of the bath Hamiltonian is made

of eigenvalues with non-positive imaginary part, otherwise the system would display instabilities, with some eigenmodes featuring an exponentially increasing number of photons with time.

8) Sometimes N means the total number of unit cells, sometimes it indicates the number of resonators. This is mess and makes the manuscript unreadable.

We thank the Reviewer for noticing this possible source of confusion. We thoroughly checked the manuscript and found that the only *typo* was in the third paragraph of the section **Quantum Optical System and Hamiltonian** (now merged with the **System–Bath Topological Correspondence** section), where we described the interaction Hamiltonian. We have now fixed this typo and highlighted the corresponding text in red in the new version of the manuscript. Regardless, whenever the number of resonators or cells needed to be specified, it indeed was. We hardly believe this small typo made the manuscript unreadable.

Besides, the current manuscript includes red texts in discussion and the reference part. It looks as a draft but not a completed manuscript.

We apologize for the confusion. The Discussion part and the references were in red as they differed from the very first draft we had submitted and they were highlighted in red as to make the distinction clear. We had no control on further processing of our submission and those parts remained red. We confirm that ours was not a draft, but rather a completed manuscript.

Because of the aforementioned problems, I can't recommend a publication of this manuscript.

We hope to have cleared all Reviewer#1's concerns and made a compelling argument to support the value of our work.

Reply to Reviewer#2

We thank the Reviewer for the careful reading and the constructive criticism. Below we provide a point-by-point reply that hopefully will clear the Reviewer's concerns.

In this paper, the authors study a class of topological photon-atom lattices and reveal an underlying connection between the topology of the photonic and the atomic Hamiltonians.

We thank the Reviewer for acknowledging the connection that we find between the topology of a photonic lattice and the one it induces on atoms judiciously coupled to it.

This compliments the authors prior work, wherein they devised methodologies to induce a non-trivial topological order in lattices of quantum emitters through interactions mediated by Hermitian and non-Hermitian photonic baths. Here, they generalize their results, to create a classification for both Hermitian and non-Hermitian cases that predicts a reversal or preservation of topological invariants, between the photonic and the atom-photon Hamiltonians, based on symmetries and dimensionality.

We kindly disagree with the Reviewer on this point.

Assuming the Reviewer is implicitly referring to Ref. [43] (which might not be necessarily the case), we would like to stress that in Ref. [43] we do not devise any methodologies to induce a non-trivial topological order in lattices of quantum emitters through interactions mediated by Hermitian and non-Hermitian photonic baths. In Ref. [43], the reversed topology between atomic and (only non-Hermitian) bath Hamiltonian is only noticed for a specific setup and explained in terms of atom-photon bound

states, with no general topological argument. Moreover, to be more precise, the specific setup considered in Ref. [43] does not fall within the one-emitter-per-resonator case category, to which the general theorems proved in the present manuscript apply. When fewer emitters than resonators are present in the system (as in Ref. [43]), we show (see Supplementary Information) that there is no general topological correspondence by providing counterexamples. Therefore, the content of Ref. [43] cannot be regarded as an instance of our general classification. In conclusion, our manuscript does not generalize Ref. [43], rather it was motivated by the findings of Ref. [43].

Assuming the Reviewer is implicitly referring to Ref. [19] (which might not be necessarily the case), this work only considers a specific (only Hermitian) bath. Also, only the topology of individual atomic bands was considered, disregarding their order in the spectrum, and the effect of topology inversion is not mentioned.

Finally, we note that none of the relevant references in waveguide quantum electrodynamics (Refs. [15-22]) devise any methodologies to induce a non-trivial topology to the quantum emitters through the (Hermitian or non-Hermitian) topology of the underlying photonic bath.

The results are interesting and indeed provide criteria for predicting the topology of certain photon-bathed atomic lattices.

We are grateful to the Reviewer for acknowledging the predictive character of our result.

However, their generality is questionable, especially considering the authors' assertion of "providing general theorems on the topology of photon-mediated interactions" as stated in their abstract. Overall, it appears that the applicability of the scheme is constrained by specific approximations, selected parameters and most importantly relies on strict rules for the way atom emitters can be arranged on the photonic resonator lattice, as revealed by the scope of cases studied.

We understand the Reviewer's concern. However, we would like to offer a different view on this matter. We start our work with our motivating question: "do atoms coupled to a photonic bath with known topological properties inherit any of those topological properties?". This question is well motivated given the state of the art of waveguide quantum electrodynamics. In order for this question (i) to make sense and (ii) to have a positive answer, certain minimal constraints (rather than approximations) need to be made. For instance,

1. the emitters' arrangement need to preserve translational invariance to easily define the topological properties of the effective atomic Hamiltonian. However, strict translational invariance can be broken with little or no harm for the topological properties of both \hat{H}_p and \hat{H}_a by adding disorder and as long as the latter preserves the symmetry class and gap of the system;
2. the emitters' frequency all need to be in resonance with the resonators: we demonstrate that this is a necessary and sufficient condition to preserve the symmetry class between photonic Hamiltonians \hat{H}_p and effective atomic Hamiltonian \hat{H}_a . In turn, having the same symmetry class is a necessary condition for the comparison between the topology of \hat{H}_p and \hat{H}_a to make sense.

In that regard, while the technical aspects are correct,

We thank the Reviewer for acknowledging the correctness of our results.

I do not think this work meets the scope of Nature Communications.

According to the manuscript, the atom emitter must conform to equation 3 which, as the authors imply, requires a large separation between the atomic frequency and the photon spectrum. Moreover, the

proof presented in section D is limited to $\omega\varepsilon=0$. It is apparent that for non-zero values the combined spectrum will lose all symmetries (minus TRS).

We thank the Reviewer for this comment.

The Reviewer is certainly correct that the proof is valid only in resonance ($\omega_e = \omega_0$), because, as said previously, the symmetry preservation between \hat{H}_p and \hat{H}_a is essential to discuss about topological classification, and having non-trivial results. The resonance condition is a necessary and sufficient condition for symmetry preservation. Although the proof require resonance, one can give up this condition and perturb the system by adding, e.g., *frequency disorder*. Some preliminary results show that the system is quite robust under this disorder (see our comment towards the end of the reply to Reviewer#2, and the new Section S3 in the Supplementary Information). So this condition, although technically necessary, may not be so strict for the robustness of the system.

We also modified the text (in red) in the first paragraph of section 2D in the Supplementary Information

“we focused on the case of $\omega_e = \omega_0 = 0$ (ω_0 is the bare resonator frequency)”

to recall the resonance condition.

Similarly, in the non-Hermitian cases the authors select $\omega\varepsilon=-iJ$, a value that sets the atom spectrum at presumably an optimal position to preserve the topology of the bath.

Actually, the value $\omega\varepsilon=-iJ$ that we set for our non-Hermitian examples is not chosen for preserving the topology. The only requirement for $\omega\varepsilon$ is that it does not belong to the photonic spectrum. The value $\omega\varepsilon=-iJ$ is chosen for mere convenience as it makes the atomic spectrum under periodic boundary conditions symmetric with respect to the imaginary axis.

I believe the authors should emphasize more the reliance of their scheme and its sensitivity to these parameters. Will the classification fail for different values of atomic frequencies?

We would say that a symmetry class reduction occurs rather than that the classification fails. That is, we should refer to a different place in the dictionary. Concretely speaking, if the atomic frequency is not on resonance, the particle-hole symmetry or chiral symmetry of the photonic Hamiltonian will no longer be inherited by the atomic Hamiltonian. Please note that this point has already been explicitly mentioned in the main text (3rd paragraph of the *Hermitian Case* subsection):

“off resonance, the following transitions of symmetry classes occur when going from H_p to H_a : AIII \rightarrow A, BDI \rightarrow AI and D \rightarrow A”

In a more complex scenario, can the emitters have different frequencies on the same lattice?

Sure, we can definitely consider this scenario, which is a promising direction for future studies.

What will happen to the topology in such case?

The situation could be very complicated. As mentioned above, this may lead to a reduction of symmetry class and correspondingly alter the classification. For example, even if $\omega\varepsilon$ is not completely random but alternating on even and odd sites in a 1D lattice, then an SSH bath can no longer imprint the topology onto the atomic chain due to the explicit breaking of chiral symmetry. Even if the symmetry is preserved or irrelevant, we expect the topology may or may not change, depending on the explicit configuration of emitter frequencies. An interesting situation could be that the frequencies fluctuate so strongly that the atomic Hamiltonian becomes gapless, yet the randomness gives rise to Anderson localization and

mobility edges, between which we still have an effective band gap. We may use some real-space topological markers, which have been developed for both Hermitian and non-Hermitian systems [see, e.g., F. Song et al., *Phys. Rev. Lett.* 123, 246801 (2019)], to characterize the topology. Moreover, inhomogeneity itself may modify the effective Hamiltonian, leading to, e.g., topological Anderson insulators [see, e.g., J. Li et al., *Phys. Rev. Lett.* 102, 136806 (2009)], which are otherwise trivial in the homogenous case.

Can it be preserved on a sublattice where $\omega\varepsilon=0$ (for the Hermitian case) if the energy separations with other atoms are generally large?

As mentioned above, this modification (not matter how small the separation is) explicitly breaks the chiral symmetry as well as the associated topology. It is like (of course, still different) a Rice-Mele model which can be obtained by directly adding an alternating potential to the SSH model. Formally speaking, we should then refer to class A, which is trivial in 1D.

In Fig. 3 it seems that an optimal range for the number of missing atoms exists. For $d=0$ the authors prove the combined lattice is trivial, while $d=4$ seems ideal. What dictates the selection of d and how is it connected to the topology? Are there general rules? What happens for $d>4$?

We thank the Reviewer for this interesting comment. The result that the topology is trivial for $d=0$, and therefore the edge states disappear, can be understood as the emitters being in resonance with the edge modes and hybridizing with them no matter how small the coupling g is. To be able to observe the bulk-edge correspondence of the atomic subsystem we either need to have a periodic photonic system, or the edge of the atomic subsystem be sufficiently far away from the edge of the photonic one. Since the edge states decay exponentially into the bulk of the photonic subsystem, already for $d=4$ we observe the expected result (and the results are very similar for any $d>4$). More quantitatively, that the minimal value for d is dictated by the localization length of the edge modes, which in turn is a function of (the inverse of) the bulk gap.

In Fig. 2, it is not clear if the photonic lattice must remain periodic (as panel 'A' and the text implies) or OBCs are also applied (as implied by the photon spectrum shown to the right).

In the caption of the Fig. 2 we specify that

“Open boundary conditions for the atomic system are obtained by removing quantum emitters, but leaving the photonic lattice unaffected (hence the bath remains translationally invariant).”

Moreover, edge states are shown for $d=8$, in which case the atomic lattice terminates 'properly' according to the OBC rules for an SSH Hamiltonian. In that case, should we expect the absence of edge states for $d=7$ (trivial topology)?

Actually, for $d=7$ there is always one edge state (the SSH model with an odd number of sites has always one edge state), localized either to the left or to the right, depending on the ratio of the two hopping strengths.

An important aspect of topological transport is its inherent resilience against disorder. To what extent these models exhibit protected transport, given that the topological properties depend on $\omega\varepsilon$ and d ?

As we mention in one of the previous answers, once d is sufficiently large the topological properties do not depend on it anymore. Experimentally, we do not expect any difficulty in fixing a suitable value of d .

Regarding the disorder, some topological phases may be stable up to a certain amount of it, depending on the AZ class they belong to. For example, in E. Prodan et al., *Phys. Rev. Lett.* 105, 115501 (2010), they study the effect of on-site disorder on the stability of a 2D topological insulator. The interesting aspect of the models we are considering is that on-site disorder (disorder in the transition frequencies) also implies disorder in the effective bath-mediated interactions. A complete analysis of the effect of disorder in all different AZ classes would constitute a work on its own, and it would also distract the reader from the main message.

Still, we added a new small section (S3) in the Supplementary Information (in red) about the robustness against disorder, in particular for the QWZ model.

While it may escape the scope of this work, it would be interesting to speculate whether analogous principles can be applied to Floquet topological baths by expanding the number of symmetry classes.

We thank the Reviewer for this interesting suggestion, which indeed supports the impact of our work.

Reply to Reviewer#3

We are grateful to Reviewer#3 for the positive assessment of our manuscript and the recommendation for publication in Nature Communications.

The manuscript titled "Hermitian and Non-Hermitian Topology from Photon-Mediated Interactions" by F. Roccati et al. presents a general theorem regarding the topology of a quantum emitter (QE) array coupled to either Hermitian or non-Hermitian photonic bath with specific topological properties.

We thank the Reviewer for acknowledging the generality of our work.

*The authors consider the total Hamiltonian of the QE-bath (photonic lattice) system and subsequently derive the effective Hamiltonian of the QEs by tracing out the degrees of freedom of the bath. This approach allows for the treatment of photonic lattices as mediators, facilitating photon-mediated interactions between QEs. The authors' findings reveal that the topology of this effective system can either be preserved or reversed across fundamental symmetry classes. This photon-mediated interaction mechanism and the approach of effective Hamiltonian were previously discussed and implemented in their previous work [43], *Optica* 9, 565 (2022). In this previous work, they discussed how non-Hermitian and topological baths could induce non-reciprocal and nearest-neighbor couplings between emitters, a key feature of the topological Hatano-Nelson array.*

We are happy to notice that the Reviewer underlines how Ref. [43] only points to – but does not include in any way - the general result of our manuscript.

*On the other hand, this bath-engineering and effective Hamiltonian approach have been implemented in non-Hermitian photonic platforms, such as parity-time (PT) symmetric and anti-PT symmetric dimmers, in both classical and quantum optics, such as *Optics Letters* 43 (21), 5371 (2018), *Nature Communications* 8, 1909 (2017), *Quantum* 7, 982 (2023). However, it's important to note that the baths in these works were generally Hermitian and lacked topological properties.*

We thank the Reviewer for noticing that our work goes beyond other previous related papers.

This manuscript bridges the gap between the topological properties of the bath and the "system" (in this case, the QE array), providing answers to the questions raised in the manuscript, particularly

regarding whether atoms coupled to a photonic bath with known topological properties inherit any of those topological characteristics.

We thank the Reviewer for acknowledging that we provide an answer to the motivating question of the work, with a specific emphasis on the case of a quantum optical system.

I found the results presented in the current manuscript to be important and interesting. This photon-mediated interactions introduced in this study has the potential to guide the experimental design of topological emitters across various platforms. Therefore, I recommend this work for publication in Nature Communications.

We are grateful to the Reviewer for the very positive assessment, for the comment on the potential of our work, and eventually for the recommendation for publication in Nature Communications.

Here are some minor points that the authors should address in a revised manuscript:

1. It would be helpful to add a schematic figure to illustrate the total atom-bath system (similar to Fig. 1b) and the effective atomic array with photon-mediated interaction (similar Fig.2a).

We thank the Reviewer for this suggestion. We added what is now Fig. 2 in the manuscript, with the caption in red, and accordingly changed the numbering of the other figures.

2. In Eq. (1), \mathfrak{h} is 1 if \hat{H}_p is Hermitian, and 0 if \hat{H}_p is non-Hermitian. Please modify the sentence “ \mathfrak{h} is 1(0) if \hat{H}_p is (non-)Hermitian” to clarify that this is only for Hermitian AZ classes and non-Hermitian AZ and AZ^\dagger classes to prevent potential confusion, as a general non-Hermitian system can continuously transition to a Hermitian one. For instance, in a Hatano-Nelson 1D array, the coupling defieerence δ can continuously change from nonzero (non-Hermitian) to zero (Hermitian).

We followed the Reviewer’s suggestion and modified accordingly. The new version reads:

“ \mathfrak{h} is 1(0) if \hat{H}_p is (non-)Hermitian and belongs to the (non-)Hermitian AZ (AZ and AZ^\dagger) classes.”

The updated text is highlighted in red in the revised manuscript.

3. Please clarify the proof of Eq. (1) can be found in supplementary section S2D .

Four lines after Eq. 1 we introduced the sentence (highlighted in red in the revised manuscript):

“The proof of Eq. 1 is provided in the Supplementary Section S2D.”

4. In the first paragraph of Non-Hermitia case, please clarify that $$, T , \dagger represent complex conjugate, transpose, and Hermitian conjugate, respectively.*

At the end of the first paragraph of **Non-Hermitian Case** we added (in red):

“where $*$, T , and \dagger represent complex conjugate, transpose, and Hermitian conjugate, respectively.”

5. Below Eq. (6), “with QEs coupled to all modes”, here “modes” should be “resonators” or unit cells.

We modified accordingly (highlighted in red).

6. Please specify I_2 in Eq. (7) is the 2-dimensional identity matrix.

After Eq. 7 we added (in red in the revised manuscript):

“where I_2 is the two-dimensional identity matrix,”

REVIEWER COMMENTS

Reviewer #2 (Remarks to the Author):

The authors have presented new arguments to support their results. I recommend emphasizing two key points in their main text for better clarity, specifically by quoting from the authors response:

“ d is sufficiently large the topological properties do not depend on it anymore” and

“The only requirement for $\omega\epsilon$ is that it does not belong to the photonic spectrum” for the NH case.

Although the last claim is initially stated in the beginning of the manuscript, it should be reemphasized in the NH section. A general observation is that the authors study too many cases, but only a specific example for each scenario. This doesn't allow the reader to understand which conditions are universally applicable, and the extent to which the parameters can vary.

Moreover, a calculation of the Kitaev sum as a real-space indicator of topology is a valid measure, and Fig. S3 reaches a conclusive argument in favor of the author's claims.

Reviewer #3 (Remarks to the Author):

The authors have addressed all my points and revised the manuscript appropriately. The addition of Supplementary Section S3 on the robustness against disorder has increased the credibility of the results in the paper. A minor suggestion is to clarify in the caption of Fig. S3 that the y-axis " v " represents the Kitaev sum to avoid confusion with the topological invariants $\nu\alpha(p)$ in the main text or the coupling v in Eq. (4). I cautiously believe that the results in this manuscript are general and applicable to various quantum systems. I hope that further related works on this topic can confirm this assertion.

Rev#2 additional comments:

The first reviewer raises some important questions. Let me first comment on his response and I will follow with my general assessment.

In general, the paper is mathematically correct and while there may be some ambiguity in the use of the term "bath", I had no issues understanding its context or the details of the models studied. The authors could emphasize even more in their text their interpretation of the term "photonic bath" and distinguish it from the "surrounding environment", which will be important when considering cavity implementations (see my following comment on decay rates).

Reviewer 1 aligns with my initial critique regarding the generality of the models and, consequently, their realistic applicability. He emphasizes two points, the importance of photon-atom resonance and whether there are restrictions in the way the photonic resonators are coupled with the atomic degrees of freedom. I believe the authors have somewhat addressed these in their response to my previous comments. Reviewer 1 emphasized the limitations of H_{int} . The authors address this in their supplementary in section S1 and figure S1. In particular they claim that the topological classification they present in the main manuscript will be valid only at resonance $\omega_{\text{e}} = \omega_0$ and in the case of 1 photonic coupler per 1 atom emitter. In general, they claim that if the coupler-to-emitter ratio is not perfect, the topological properties can still transfer into the atomic degrees of freedom, but without generality regarding the topological classification (as in, Equation 1 will not be valid in the general case). In this regard, the new comment of reviewer 1 on the new Fig. 2 is correct and the authors should address it.

My initial review (of potential rejection) was based on similar points, but following my second review, I now remain more neutral. I understand the limitations of the model, but I agree that it can present an effective methodology to design topological hybrid models in a new way. Nonetheless, the new point 5 of reviewer 1 raises some concerns. When it comes to a realistic implementation (usually the typical designs involve cavities), the quality factor and by extend the cavity decays are important. This is especially true when applying approximations to the models that result in strict conditions (such as the condition $\omega_{\text{e}} = \omega_0$, which the reviewers reemphasized is important to the Hermitian case in response to my previous review). I have looked into previous work from the authors (such as ref 43) where they briefly discuss various alternative implementations like superconducting circuits, but with no clear answer to this concern. In this regard the authors must provide a more concise and formal answer to this point if they wish to pursue a publication in Ncomms.

Reviewer #1 (Remarks to the Author):

The report includes many mathematic symbols and equations. Thus, I prepared it with latex and generated a pdf file. Please see the attached report.

Generally speaking the revised manuscript has been improved to some degree. However, the authors still leave a lot of issues to be addressed and only complain about my previous comments in reply. Below are some examples but not all.

1. In physics and quantum optics, we typically refer “bath” to the quantum vacuum fluctuations in open space when we consider a system consisting of atoms and optical resonators. Such bath includes many distinct modes. This basic knowledge can be found in many textbooks of optics and atomic physics. If one talks about a bath with different meaning, the meaning of the bath needs to be clarified. After reading the original and revised manuscripts many times, I can guess the authors refer to the bath as the optical modes of the photonic resonators/lattice. If so, the authors should make it clear at the beginning of the model and clarify that this is an artificially engineered ”bath”. Otherwise, the manuscript is very confusing and misleading. Moreover, the authors often use the term “photonic lattice” when they mention the “bath”, e.g. in the caption of Fig. 1(B). This is confusing again. The modes in photonic lattice/resonators is well defined. By designing a specific topological photonic circuit/lattice, the bath can be engineered to have unique features. However, the features are normally crucially dependent on the structure of photonic lattice. It is hard to give a general conclusion.
2. The authors should justify why and in what case the couplings of atoms and the optical resonator to “their own baths” (open space) can be neglected, which normally cause decays. The decay can be very crucial for the quantum system dynamics and features. In cavity QED systems, the decay plays essential roles.
3. The authors consider the case of “an emitter arrangement with periodicity equal to or greater than that of the photonic lattice”. This is conflict with the interaction Hamiltonian H_{int} when two periodicities are not equal.

Let’s write down the interaction Hamiltonian below for discussion.

$$\hat{H}_{\text{int}} = g \sum_{n=1}^{N_c} \sum_{s \in \mathcal{C}} (\hat{\sigma}_{ns}^\dagger \hat{a}_{ns} + H.c.) , \quad (1)$$

where $\mathcal{C} \subseteq \{1, \dots, N_b\}$ is the set of sublattices coupled to the QEs. In this form, there is at least one emitter coupling to the resonators in each unit cell because S takes some numbers from \mathcal{C} , simply speaking $S \geq 1$, for each n . In this case, the period of emitters equals to that of the photonic lattice. For non-equal periodicities, some emitters decouple from the photonic lattice for some n . Then, the sum over n from 1 to N_c is wrong. The coupling rate g should not be outside the sum as a global constant. For some emitters, it should be zero.

4. The current form of Hamiltonian \hat{H}_{int} is very confusing. It means that each emitter only couples to one resonator in the sublattice. It can’t describe the system shown in Fig. 3A. In this configuration, two neighbouring emitters couples to each other. It requires the coupling form $\sigma_j^\dagger \sigma_j + 1$.
5. The equation 3 is problematic. It describes the optical Stark shift to the emitters. To adiabatically eliminate the resonators, the resonator needs to be off resonance with the emitters, requiring $\omega_0 - \omega_e$ much larger than the decay rates of the resonators. So, the assumption of resonance condition, i.e. $\omega_0 = \omega_e$, is not valid. Otherwise, the equation 3 divergings. The authors can check this equation with the system shown in Fig.1(B).
6. The sentence “It consists of N_e two-level quantum emitters (QEs)” is unclear. Does it mean the total number of emitters in the system or within a unit cell?

7. In equation S3, the Bloch Hamiltonian for a 1D photonic lattice use the Bosonic operators σ_x , σ_y and σ_z . These operators are used for interpretation of the interaction in the paragraph following the equation S3. However, the operators σ has already been used for emitters. Again, this confuses readers.

There are so many physical, mathematic and technical issues in the manuscript. Therefore, I don't think this manuscript deserves a publication on Nature Communications.

Reply to all the Reviewers

Dear Reviewers,

We have carefully considered all your concerns, to which we provide an extensive point-by-point reply here below (with quoted text in italic and our replies in blue) and accordingly made a major revision of our manuscript. All the new/modified text in the manuscript is in red.

Considering our response and our additional work (cf. new version of our manuscript attached), we hope that our work could now be accepted for publication in Nature Communications.

Sincerely,

Federico Roccati, Miguel Bello, Zongping Gong, Masahito Ueda, Francesco Ciccarello, Aurelia Chenu, Angelo Carollo

Reply to Reviewer#1

Before providing a point-by-point reply, we would like to point out that part of our major revision of the manuscript (mainly motivated by the Reviewer's points 1, 2 and 5) consisted in the restructuring of Figs. 1 and 2 of the previous version. These have now become Fig. 1 and Table 1. The former illustrates our setup, highlighting the underlying hypotheses, while the latter summarizes our main result.

- 1. In physics and quantum optics, we typically refer "bath" to the quantum vacuum fluctuations in open space when we consider a system consisting of atoms and optical resonators. Such bath includes many distinct modes. This basic knowledge can be found in many textbooks of optics and atomic physics. If one talks about a bath with different meaning, the meaning of the bath needs to be clarified. After reading the original and revised manuscripts many times, I can guess the authors refer to the bath as the optical modes of the photonic resonators/lattice. If so, the authors should make it clear at the beginning of the model and clarify that this is an artificially engineered "bath". Otherwise, the manuscript is very confusing and misleading. Moreover, the authors often use the term "photonic lattice" when they mention the "bath", e.g. in the caption of Fig. 1(B). This is confusing again.*

We thank the Reviewer for raising this possibly confusing point. To benefit clarity, in the revised manuscript we added the sentence

“In these setups, the photonic lattice acts as an artificially *engineered bath, or environment*, for the quantum emitters as their decay rate into the lattice guided modes is larger than their decay rate in free space (56, 57).”

with the new references (56) [D. E. Chang, et al., *Rev. Mod. Phys.* **90**, 031002], and (57) [A. S. Sheremet et al., *Rev. Mod. Phys.* **95**, 015002 (2023)], at the end of the 2nd paragraph of the **INTRODUCTION** section.

In addition, we introduced a new sentence at the beginning of the 2nd paragraph of the subsection **System–Bath Topological Correspondence**, namely

“In the following we detail the type of systems to which our theory is applicable. To be precise, we distinguish between photonic modes of the engineered bath and those of the surrounding space. Coupling to the latter can be modelled through non-Hermitian Hamiltonians, both for the engineered bath and the quantum emitters. We may also neglect the surrounding environmental modes and use Hermitian Hamiltonians instead, assuming that the emitters couple more strongly to the engineered photonic bath modes (37, 58). Thus, the whole system...”

and correspondingly added the reference (58) [A. González-Tudela and J. I. Cirac, Markovian and non-Markovian dynamics of quantum emitters coupled to two-dimensional structured reservoirs. *Phys. Rev. A* **96**, 043811 (2017)].

Furthermore, in the paragraph starting with “In the Markovian regime...” we replaced

“(smaller than the spectral distance between ω_e and the bands of the bath), the bath then induces effective coherent interactions between the emitters”

with

“(smaller than the spectral distance between ω_e and the photonic bands), the photonic lattice then induces effective coherent interactions between the emitters”.

The modes in photonic lattice/resonators is well defined. By designing a specific topological photonic circuit/lattice, the bath can be engineered to have unique features. However, the features are normally crucially dependent on the structure of photonic lattice. It is hard to give a general conclusion.

We thank the Reviewer for stressing this point which is actually crucial. Through a proper design of the hopping rates of the photonic lattice/bath, certain unique topological features can emerge. These features do indeed depend on the structure of the photonic lattice. Given a (general) photonic lattice with topological features, we study if and how these are transferred to a set of quantum emitters that are coupled to such a photonic lattice. In this sense, fixing the topological properties of the photonic lattice (which is *not* the same as fixing the couplings between the resonators; different photonic lattices, with a different design of the hopping rates, could still have the same topological properties), our conclusions are general.

2. *The authors should justify why and in what case the couplings of atoms and the optical resonator to “their own baths” (open space) can be neglected, which normally cause decays. The decay can*

be very crucial for the quantum system dynamics and features. In cavity QED systems, the decay plays essential roles.

We thank the Reviewer for pointing out this crucial issue. We believe this question by the Reviewer to be twofold.

On the one hand, there is the question of why and how the photonic lattice (our engineered bath) can be considered isolated from the surrounding environment. Of course, no real quantum system can be fully isolated from the surrounding environment, not to mention our will to measure and interact with it. However, in certain parameter regimes, which depend on the experimental implementations, a photonic lattice can be considered isolated from the surrounding environment, see, *e.g.*, Ref. (56). For instance, in a cavity QED implementation, the figure of merit is the cavity quality factor $Q = \omega/\gamma$, where ω is the resonant frequency of the cavity mode and γ the dissipation rate. As the Reviewer notices, “*the decay can be very crucial for the quantum system dynamics and features*”. Indeed a photonic lattice can be engineered to be non-Hermitian (*i.e.*, dissipative, that is, not-isolated from its surrounding environment), which is important for our result, in various ways. For example, (i) in a cavity QED implementation some of the cavities (in a translationally invariant fashion) can be intentionally fabricated to have a low quality factor, or (ii) in circuit QED, where each resonator is a superconducting circuit, collective dissipation can be achieved by coupling the resonators of the lattice to a transmission line [see, *e.g.*, A. Blais et al., *Rev. Mod. Phys.* **93**, 025005 (2021)].

On the other hand, there is the question of why and how a photonic lattice can be considered as a bath for quantum emitters coupled to it, or, alternatively, why their decay into free space can be neglected. The possibility of coupling emitters to structured photonic arrays, neglecting the decay in free space, is at the core of quantum nanophotonics, cf., *e.g.*, Ref. (56). It is of course true that the quantum emitters will always have a residual decay in free space. However, under many circumstances their decay rate into the guided modes of the photonic lattice (Γ_g) can be made larger than the decay rate in free space (Γ). The specific value of Γ_g/Γ depends on the specific experimental implementation. For example, for atoms (*i.e.*, the quantum emitters) near a nanofiber (*i.e.*, the photonic engineered bath) the ratio Γ_g/Γ is enhanced by reducing the distance between the atom and the fiber [see, *e.g.*, F. Le Kien et al., *Phys. Rev. A* **72**, 032509 (2005)].

A more refined analysis on this point can be found in Ref. (57) and we believe that such a long discussion would go beyond the scope of the manuscript. Nevertheless, we added the sentence mentioned in the reply to point 1 (and pointed to relevant references), and a new paragraph in red in the **DISCUSSION** section.

3. *The authors consider the case of “an emitter arrangement with periodicity equal to or greater than that of the photonic lattice”. This is conflict with the interaction Hamiltonian \hat{H}_{int} when two periodicities are not equal.*

Let's write down the interaction Hamiltonian below for discussion.

$$\hat{H}_{int} = g \sum_{n=1}^{N_c} \sum_{s \in \mathcal{C}} (\hat{\sigma}_{ns}^\dagger \hat{a}_{ns} + H.c.),$$

where $\mathcal{C} \subseteq \{1, \dots, N_b\}$ is the set of sublattices coupled to the QEs. In this form, there is at least one emitter coupling to the resonators in each unit cell because S takes some numbers from \mathcal{C} , simply speaking $S \geq 1$, for each n . In this case, the period of emitters equals to that of the photonic lattice. For non-equal periodicities, some emitters decouple from the photonic lattice for some n . Then, the

sum over n from 1 to N_c is wrong. The coupling rate g should not be outside the sum as a global constant. For some emitters, it should be zero.

We thank the Reviewer for this comment. Please note that n labels a unit cell of the *entire* system (emitters + photonic lattice) rather than a specific site, and that N_c denotes the total number of unit cells. We agree that putting g into the sum is more comprehensible and transparent than specifying a subset \mathcal{C} . In order to fully clarify the interaction Hamiltonian, we have modified the sentence where we introduce the interaction Hamiltonian to:

“... $\hat{H}_{\text{int}} = \sum_{n=1}^{N_c} \sum_{s=1}^{N_b} g_s (\hat{\sigma}_{ns}^\dagger \hat{a}_{ns} + H.c.)$. Here, \hat{a}_{ns} is the real-space annihilation operator of the resonator located in the n th unit cell, belonging to the s -sublattice ($s = 1, \dots, N_b$), and N_c is the total number of unit cells. The atomic operator, $\hat{\sigma}_{ns}$, in \hat{H}_{int} has two indices to specify the resonator to which it is coupled. The coupling strength g_s satisfies $g_s = g$ if a QE is coupled to the resonator \hat{a}_{ns} , and is set to zero otherwise. Note that g_s is independent on the cell index n , ensuring translational invariance.”

Accordingly, before Equation 1, we have modified the sentence

“In addition, when the system and bath have the same number of degrees of freedom...”

to

“In addition, when the system and bath have the same number of degrees of freedom (*i.e.*, $g_s = g$ for all s) ...”

and modified the sentence after Equation 1

“The topological correspondence we found holds in the case of one emitter per resonator, namely $\mathcal{C} = \{1, \dots, N_b\}$. Violations of this correspondence in the fewer-emitters-than-resonators case ($\mathcal{C} \subset \{1, \dots, N_b\}$) are discussed in the supplementary section S1.”

to

“The topological correspondence we found holds in the case of one emitter per resonator, namely $g_s = g$ for all s . In the case where there are fewer emitters than resonators ($g_s = 0$ for some s), this general correspondence does not hold anymore, as discussed in the supplementary section S1.”

4. *The current form of Hamiltonian \hat{H}_{int} is very confusing. It means that each emitter only couples to one resonator in the sublattice.*

We hope that our reply to point 3 has now dissipated this concern.

It can't describe the system shown in Fig. 3A. In this configuration, two neighbouring emitters couples to each other. It requires the coupling form $\hat{\sigma}_j^\dagger \hat{\sigma}_j + 1$.

We thank the Reviewer for raising this potentially misleading point. In the total Hamiltonian $\hat{H} = \hat{H}_e + \hat{H}_p + \hat{H}_{\text{int}}$ no direct coupling between the QEs is present. The couplings between the QEs only emerge when tracing out the photonic lattice/bath. Thus, following the scheme of Fig. 2 (now Fig. 1), we have revised Fig. 3 (now Fig. 2), and we hope this fully dissipates the concern.

5. *The equation 3 is problematic. It describes the optical Stark shift to the emitters. To adiabatically eliminate the resonators, the resonator needs to be off resonance with the emitters, requiring $\omega_0 - \omega_e$ much larger than the decay rates of the resonators. So, the assumption of resonance condition, i.e. $\omega_0 = \omega_e$, is not valid. Otherwise, the equation 3 divergences. The authors can check this equation with the system shown in Fig.1(B).*

We thank the Reviewer for the comment. We confirm that the condition $\omega_0 = \omega_e$ is fully consistent with adiabatic elimination and no divergence arises. Indeed, notice that here the field's characteristic frequency (which emitters must be detuned from) *is not* ω_0 (as implicitly assumed in the Reviewer's above comment) since resonators in our system are mutually *coupled* which results in photonic bands and *bandgaps*. Depending on the lattice structure, it is thus perfectly possible [see, e.g., our example about QEs couple to an SSH photonic lattice, Fig. 3 (now Fig. 2); or Ref. (16)] to have a bandgap centered just at ω_0 , so that adiabatic elimination can in particular occur for $\omega_0 = \omega_e$ (under weak coupling).

Indeed, we make well clear in the beginning the assumption that " ω_e lies within a photonic bandgap" (see **System–Bath Topological Correspondence** section), and we stress this throughout the manuscript (e.g., before the (NH) *Topological Reversal* subsection).

6. *The sentence "It consists of N_e two-level quantum emitters (QEs)" is unclear. Does it mean the total number of emitters in the system or within a unit cell?*

We thank the Reviewer for noticing this point. We have now modified this sentence to:

"It consists of two-level quantum emitters, N_e in total,..."

7. *In equation S3, the Bloch Hamiltonian for a 1D photonic lattice use the Bosonic operators σ_x , σ_y and σ_z . These operators are used for interpretation of the interaction in the paragraph following the equation S3. However, the operators σ has already been used for emitters. Again, this confuses readers.*

In order to avoid any confusion, we replaced throughout the manuscript the labels σ_x , σ_y and σ_z , whenever they are used as Pauli matrices to represent Bloch Hamiltonians, with τ_x , τ_y and τ_z , respectively.

We hope to have cleared all Reviewer#1's concerns and made a compelling argument to support the value of our work.

Reply to Reviewer#2

The authors have presented new arguments to support their results.

We thank Reviewer#2 for this positive statement.

I recommend emphasizing two key points in their main text for better clarity, specifically by quoting from the authors response: "d is sufficiently large the topological properties do not depend on it anymore" and

“The only requirement for $\omega\varepsilon$ is that it does not belong to the photonic spectrum” for the NH case. Although the last claim is initially stated in the beginning of the manuscript, it should be reemphasized in the NH section.

Following Reviewer#2’s suggestions, we have added the sentences

“We emphasize that when d is sufficiently large the topological properties do not depend on it anymore.”

in the caption of Fig. 4 (now Fig. 3) in the main text, and

“We recall that the only requirement on ω_e is that it does not belong to the photonic spectrum.”

before the (NH) *Topological Reversal* subsection.

A general observation is that the authors study too many cases, but only a specific example for each scenario. This doesn't allow the reader to understand which conditions are universally applicable, and the extent to which the parameters can vary.

We thank Reviewer#2 for this comment. To help and guide the reader we have added the sentence

“We now proceed illustrating one example of topological preservation and reversal in each of the four possible cases (Hermitian/non-Hermitian and odd/even spatial dimension). The proofs are detailed in the supplementary section S2D.”

right before the **Topological Preservation and Reversal** section. Now it is clear that the number of examples is *minimal* rather than “too many” – it covers all the four variations (Hermitian or non-Hermitian, odd or even dimension) of our main results. Moreover, we note that our choices of the models are the most typical representatives in the periodic tables for Hermitian and non-Hermitian AZ classes, just like Ref. (34) where similar specific models are used to demonstrate the general Hermitian-non-Hermitian correspondence. Finally, those readers who want to understand the detailed derivations will be naturally guided to the Supplementary Materials.

Moreover, a calculation of the Kitaev sum as a real-space indicator of topology is a valid measure, and Fig. S3 reaches a conclusive argument in favor of the author’s claims.

We thank Reviewer#2 for recognizing that our further study strengthens our claims.

Reply to Reviewer#3

The addition of Supplementary Section S3 on the robustness against disorder has increased the credibility of the results in the paper.

We thank the Reviewer for this positive comment.

A minor suggestion is to clarify in the caption of Fig. S3 that the y-axis “ v ” represents the Kitaev sum to avoid confusion with the topological invariants $\nu_a(p)$ in the main text or the coupling v in Eq. (4).

We have added the following sentence to the caption of Figure S3:

“We note that in the y-axis, " ν " represents the Kitaev sum (not any of the topological invariants mentioned in the main text).”

I cautiously believe that the results in this manuscript are general and applicable to various quantum systems. I hope that further related works on this topic can confirm this assertion.

We are grateful to the Reviewer for recognizing the potential impact of our work.

REVIEWERS' COMMENTS

Reviewer #1 (Remarks to the Author):

The manuscript has been greatly improved in comparison with the previous version. If

1) In the reply, the authors claim that no divergence arises in Eq. (3). However, what is form of $H_p(K)$ and the value $\omega_e - H_p(K)$ in Eq. (3) at resonance $\omega_e = \omega_0$. ω_0 is not the field's characteristic frequency but the bare resonator frequency. $H_p(K)$ should use the frequency ω_0 but not the field's.

2) The interaction Hamiltonian changes after the g moves into the notation sum. It is not a constant value for all unit sites now (g_s can be g or 0). Then, the model changes, the results based this model also changes? Is it correct to use the results from previous model as it treats g_s as a global value g ?

I am still not completely satisfied with the response and the revised manuscript after three-round review.

Reviewer #2 (Remarks to the Author):

The authors made a number of improvements to the presentation of their work. I am satisfied with the changes. I recommend publication of their manuscript.

Reply to the Reviewer#1

The quoted text is in italic, while our replies are in blue.

The manuscript has been greatly improved in comparison with the previous version.

We thank the Reviewer for acknowledging that our manuscript has greatly improved.

If 1) In the reply, the authors claim that no divergence arises in Eq. (3). However, what is form of $H_p(K)$ and the value $\omega_e - H_p(K)$ in Eq. (3) at resonance $\omega_e = \omega_0$. ω_0 is not the field's characteristic frequency but the bare resonator frequency. $H_p(K)$ should use the frequency ω_0 but not the field's.

We have addressed this point in our previous reply. Still, we would like to reply to this specific point.

We confirm that the condition $\omega_0 = \omega_e$ is fully consistent with adiabatic elimination and no divergence arises. Indeed, notice that the field's characteristic frequency (which emitters must be detuned from) is *not* ω_0 (as implicitly assumed in the Reviewer's comment) since resonators in our system are mutually *coupled* which results in photonic bands and *bandgaps*. Depending on the lattice structure, it is thus perfectly possible [see, e.g., our example about QEs couple to an SSH photonic lattice, Fig. 2; or Ref. (16)] to have a bandgap centered just at ω_0 , so that adiabatic elimination can in particular occur for $\omega_0 = \omega_e$ (under weak coupling).

2) The interaction Hamiltonian changes after the g moves into the notation sum. It is not a constant value for all unit sites now (g_s can be g or 0). Then, the model changes, the results based this model also changes? Is it correct to use the results from previous model as it treats g_s as a global value g ?

We have addressed this point in our previous reply Still, we would like to reply again to this specific point.

We included the atom-light coupling strength g_s inside the sum, therefore making it generally dependent on the sublattice index s (but not on the cell index n , so to preserve translational invariance). For the general results we discuss in the main text, each resonator is coupled to a quantum emitter, so that $g_s = g$ for all s . In the supplementary information, we discuss the violation of the topological correspondence happening when $g_s = 0$ for some s .